# LEARNING INVARIANCES FOR CAUSAL ABSTRACTION INFERENCE

## ABSTRACT

Causal abstraction inference is the task of inferring causal effects from limited data by first mapping the complicated low-level data (e.g., pixels) into a simpler high-level space (e.g., image representation) before performing causal inferences on the high-level. A major restriction in this task is known as the abstract invariance condition (AIC), which forces high-level representations to retain all information from the low-level data to prevent any ambiguity in high-level inference. In this work, we provide the first approach that can learn low-dimensional high-level representations that satisfy the strictest form of the AIC without weakening the allowable causal inferences. We show how the concept of invariances, such as rotational invariance in image data, is related to causal abstractions and how they can be used to learn lower dimensional representations using out-of-the-box invariance learning tools such as contrastive learning. Finally, we demonstrate our findings empirically, including in a high-dimensional image setting.

## 1 INTRODUCTION

Causality is a key component of human reasoning, allowing one to plan a course of action, to determine blame and responsibility, and to generalize across changing environments. A key insight from both causality and the philosophy of science is that effective reasoning often involves abstraction – the process of simplifying a complex system by ignoring details deemed irrelevant to the task. In this context, "irrelevant details" typically refer to certain transformations that leave important aspects of the system unchanged, called *invariances* in the machine learning literature. For example, humans interpret the object in the television as a "dog" rather than a collection of pixels, and this interpretation does not change whether the pixels are rotated, flipped, or cropped. The pixels are abstracted to the concept of a "dog", and it is invariant to transformations such as rotation. Invariances, when studied under the lens of causal abstractions, can therefore be a powerful tool for advancing AI systems.

Modern AI systems are often studied under the foundation of generative modeling. Deep generative models have shown impressive results in many practical tasks such as image generation (Brown et al., 2020), text generation (Ramesh et al., 2021), and style transfer (Gatys et al., 2015). Causal inference is typically studied under the semantics of structural causal models (SCMs) (Pearl, 2000), which are generative models that represent reality with a collection of mechanisms and exogenous noise. Each SCM induces a collection of distributions that can be categorized into three successively more descriptive layers known as the Pearl Causal Hierarchy (PCH) (Pearl and Mackenzie, 2018; Bareinboim et al., 2022). These layers refer to the observational ($\mathcal{L}_1$), interventional ($\mathcal{L}_2$), and counterfactual ($\mathcal{L}_3$) distributions. While traditional generative modeling focuses on a single distribution (usually the observational distribution from $\mathcal{L}_1$), causal generative modeling is an emerging field that aims to extend the capabilities of generative modeling to higher layers of the hierarchy. It has been shown that, given the proper causal constraints, causal generative models are capable of identifying, estimating, and sampling causal effects, trained on limited available data such as observational data (Kocaoglu et al., 2018; Xia et al., 2021; 2023; Rahman and Kocaoglu, 2024).

Formal studies of causal abstractions typically aim to compare a low-level model $\mathcal{M}_L$ with a high-level counterpart $\mathcal{M}_H$ through an abstraction function $\tau$ that maps low-level variables $\mathbf{V}_L$ to high-level variables $\mathbf{V}_H$. Semantic definitions such as exact transformations and $\tau$-abstractions establish key properties expected of abstractions such as the commutativity of interventions and abstractions (Rubenstein et al., 2017; Beckers and Halpern, 2019; Beckers et al., 2019; Geiger et al., 2023a).

These properties have been useful in the explainable AI domain, where a high-level causal model is hypothesized to explain a black-box model such as a neural network, and an abstraction function $\tau$ is learned to test this hypothesis by seeing how well the function satisfies these important properties (Geiger et al., 2023b; Massidda et al., 2023; Zennaro et al., 2023; Felekis et al., 2024). Separately, constructive abstractions have been useful for an emerging field of study called *causal abstraction inference*, the main focus of this work. The concept is shown in Fig. 1. While many established abstraction definitions focus on comparing SCMs $\mathcal{M}_L$ and $\mathcal{M}_H$, recent work has decomposed abstraction analysis into individual distributions of the PCH, which allows one to perform causal inferences in the high-level space given limited data from the low-level space (Xia and Bareinboim, 2024; 2025). This allows one to perform high-dimensional causal inferences tractably by first converting the data to a high-level abstract space (akin to representation learning).

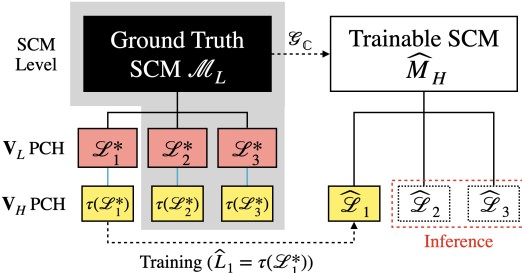

Figure 1: An illustration of the causal abstraction inference problem. The true model is a low-level model $\mathcal{M}_L$ which generates distributions of the PCH over $\mathbf{V}_L$. $\mathbf{V}_L$ is connected to its high-level counterpart $\mathbf{V}_H$ through $\tau$. In practice, $\mathcal{M}_L$ and data from interventional ($\mathcal{L}_2$) and counterfactual ($\mathcal{L}_3$) may not be available (in gray). The goal is to construct an SCM $\widehat{M}_H$ on the high-level space, apply causal assumptions in the form of constraints ($\mathcal{G}_\mathbb{C}$), train it on available observational data ($\mathcal{L}_1$), and then use it to infer $\mathcal{L}_2$ and $\mathcal{L}_3$ queries.

One particularly challenging restriction in the causal abstraction inference task that is not present in typical noncausal representation learning problems is known as the *abstract invariance condition* (AIC). The AIC states, informally, that to preserve correctness in high-level causal inferences, a high-level representation must disambiguate values that have different causal effects on downstream variables. This is illustrated in Fig. 2. A classic instance of this phenomenon is the study of the effects of cholesterol on heart disease (Spirtes and Scheines, 2004). There are two types of cholesterol, HDL and LDL, that both affect heart disease rates, so scientists may be tempted to abstract them together as total cholesterol. However, deeper analysis shows that HDL lowers the risk of heart disease while LDL raises it. Abstracting them together as total cholesterol leaves the analysis ambiguous, as one would not be able to assess the risk of heart disease without knowing whether the total cholesterol consists more of HDL or LDL cholesterol.

Since the true structural model is typically not available in most practical settings, it is generally impossible to verify that the AIC holds, leading to severe constraints on the types of representations that can be learned. Xia and Bareinboim (2024) accommodates this issue by enforcing bijectivity in learned representations through an autoencoder structure, but this approach suffers from a lack of dimensionality reduction, which is one of the main purposes of representation learning. Chalupka et al. (2015) explores a weaker version of the AIC that is verifiable by data, but this implies weaker inferences. Xia and Bareinboim (2025) generalizes the abstraction framework to show that high-level inferences under AIC violations can be corrected by interpreting them as soft interventions on the low-level model, but this requires additional assumptions to specify the form of the soft interventions and leaves fewer identifiable results.

In this work, we present an approach that leverages the availability of invariance information to learn representations that (1) satisfy the most fundamental form of the AIC, (2) allow for dimensionality reduction, and (3) make no additional assumptions (other than invariance information) without sacrificing inferential power. More specifically, in Sec. 2, we formally define invariances in the context of causal models and prove that they can be used to generate low-dimensional representations that still satisfy the AIC. Importantly, this allows for out-of-the-box techniques for invariance learning used in noncausal contexts to learn representations in causal models. In Sec. 3, we show how to use one such popular technique, contrastive learning (Chen et al., 2020), to accomplish this in practice. We then empirically demonstrate the strength of the learned representations in Sec. 4 before concluding our findings in Sec. 5. Due to space constraints, proofs can be found in App. A.

## 1.1 PRELIMINARIES

This section introduces the notation and definitions used throughout the paper. We use uppercase letters ($X$) to denote random variables and lowercase letters ($x$) to denote corresponding values. Similarly, bold uppercase ($\mathbf{X}$) and lowercase ($\mathbf{x}$) letters denote sets of random variables and values respectively. We use $\mathcal{D}_X$ to denote the domain of $X$ and $\mathcal{D}_{\mathbf{X}} = \mathcal{D}_{X_1} \times \cdots \times \mathcal{D}_{X_k}$ for the domain of $\mathbf{X} = \{X_1, \ldots, X_k\}$. We denote $P(\mathbf{X} = \mathbf{x})$ (often shortened to $P(\mathbf{x})$) as the probability of $\mathbf{X}$ taking the values $\mathbf{x}$ under the distribution $P(\mathbf{X})$. We use the notation $\mathbf{z}[\mathbf{W}]$ to indicate the values of $\mathbf{z}$ restricted to variables in $\mathbf{Z} \cap \mathbf{W}$. We utilize the basic semantic framework of structural causal models (SCMs) (Pearl, 2000), following the presentation in Bareinboim et al. (2022).

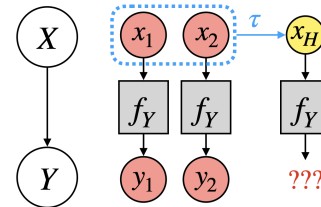

Figure 2: An illustration of an AIC violation. Note that $X$ causes $Y$, and $x_1$ and $x_2$ are different values of $X$ that provide different outputs in $Y$. If they are abstracted into the same high-level value $x_H$, then the behavior of $f_Y$ is ambiguous on the input of $x_H$.

**Definition 1** (Structural Causal Model (SCM)). An SCM $\mathcal{M}$ is a 4-tuple $\langle \mathbf{U}, \mathbf{V}, \mathcal{F}, P(\mathbf{U}) \rangle$, where $\mathbf{U}$ is a set of exogenous variables (or "latents") that are determined by factors outside the model; $\mathbf{V}$ is a set $\{V_1, V_2, \ldots, V_n\}$ of (endogenous) variables of interest that are determined by other variables in the model – that is, in $\mathbf{U} \cup \mathbf{V}$; $\mathcal{F}$ is a set of functions $\{f_{V_1}, f_{V_2}, \ldots, f_{V_n}\}$ such that each $f_{V_i}$ is a mapping from (the respective domains of) $\mathbf{U}_{V_i} \cup \mathbf{Pa}_{V_i}$ to $V_i$, where $\mathbf{U}_{V_i} \subseteq \mathbf{U}$, $\mathbf{Pa}_{V_i} \subseteq \mathbf{V} \setminus V_i$, and the entire set $\mathcal{F}$ forms a mapping from $\mathbf{U}$ to $\mathbf{V}$. That is, for $i = 1, \ldots, n$, each $f_{V_i} \in \mathcal{F}$ is such that $v_i \leftarrow f_{V_i}(\mathbf{pa}_{V_i}, \mathbf{u}_{V_i})$; and $P(\mathbf{U})$ is a probability function defined over the domain of $\mathbf{U}$. ∎

Each SCM induces distributions from the 3 layers of the PCH. This work is general to all three layers, but for clarity, we define the set of layer 2 distributions as follows.

**Definition 2** (Layer 2 Valuation (Bareinboim et al., 2022, Def. 5)). An SCM $\mathcal{M} = \langle \mathbf{U}, \mathbf{V}, \mathcal{F}, P(\mathbf{U}) \rangle$ induces a family of joint distributions over $\mathbf{V}$, one for each intervention $\mathbf{x}$. For each $\mathbf{Y} \subseteq \mathbf{V}$, $P^{\mathcal{M}}(\mathbf{y} \mid do(\mathbf{x})) = \int_{\mathcal{D}_{\mathbf{U}}} \mathbb{1}\{\mathbf{Y}_{\mathbf{x}}(\mathbf{u}) = \mathbf{y}\} dP(\mathbf{u})$, where $\mathbf{Y}_{\mathbf{x}}(\mathbf{u})$ is the solution for $\mathbf{Y}$ in the submodel $\mathcal{M}_{\mathbf{x}} = \langle \mathbf{U}, \mathbf{V}, \mathcal{F}_{\mathbf{x}}, P(\mathbf{U}) \rangle$, where $\mathcal{F}_{\mathbf{x}} := \{f_V : V \in \mathbf{V} \setminus \mathbf{X}\} \cup \{f_X \leftarrow x : X \in \mathbf{X}\}$. ∎

$\mathcal{L}_2$ is the set of all such distributions, and $\mathcal{L}_1$ is the subset where $\mathbf{X} = \emptyset$. $\mathcal{L}_3$ is defined in App. A.2. The theory of causal abstractions developed in this paper build on the foundations of constructive abstraction functions, under which individual distributions of the PCH are well-defined between low and high-level models.

**Definition 3** (Inter/Intravariable Clusterings (Xia and Bareinboim, 2024, Def. 5)). Let $\mathcal{M}$ be an SCM over $\mathbf{V}$. A set $\mathbb{C}$ is said to be an intervariable clustering of $\mathbf{V}$ if $\mathbb{C} = \{\mathbf{C}_1, \mathbf{C}_2, \ldots \mathbf{C}_n\}$ is a partition of a subset of $\mathbf{V}$. $\mathbb{C}$ is further considered admissible w.r.t. $\mathcal{M}$ if for any $\mathbf{C}_i \in \mathbb{C}$ and any $V \in \mathbf{C}_i$, no descendent of $V$ outside of $\mathbf{C}_i$ is an ancestor of any variable in $\mathbf{C}_i$. That is, there exists a topological ordering of the clusters of $\mathbb{C}$ relative to the functions of $\mathcal{M}$. A set $\mathbb{D}$ is said to be an intravariable clustering of variables $\mathbf{V}$ w.r.t. $\mathbb{C}$ if $\mathbb{D} = \{\mathbb{D}_{\mathbf{C}_i} : \mathbf{C}_i \in \mathbb{C}\}$, where $\mathbb{D}_{\mathbf{C}_i} = \{\mathcal{D}^1_{\mathbf{C}_i}, \mathcal{D}^2_{\mathbf{C}_i}, \ldots, \mathcal{D}^{m_i}_{\mathbf{C}_i}\}$ is a partition (of size $m_i$) of the domains of the variables in $\mathbf{C}_i$, $\mathcal{D}_{\mathbf{C}_i}$. ∎

**Definition 4** (Constructive Abstraction Function (Xia and Bareinboim, 2024, Def. 6)). A function $\tau : \mathcal{D}_{\mathbf{V}_L} \to \mathcal{D}_{\mathbf{V}_H}$ is said to be a constructive abstraction function w.r.t. inter/intravariable clusters $\mathbb{C}$ and $\mathbb{D}$ iff $\tau$ is composed of subfunctions $\tau_{\mathbf{C}_i}$ for each $\mathbf{C}_i \in \mathbb{C}$ such that $\mathbf{v}_H = \tau(\mathbf{v}_L) = (\tau_{\mathbf{C}_i}(\mathbf{c}_i) : \mathbf{C}_i \in \mathbb{C})$, where $\tau_{\mathbf{C}_i}(\mathbf{c}_i) = v^j_{H,i}$ if and only if $\mathbf{c}_i \in \mathcal{D}^j_{\mathbf{C}_i}$. ∎

In this work, we leverage causal diagrams (often denoted as $\mathcal{G}$) and their corresponding cluster causal diagrams (C-DAGs) (denoted as $\mathcal{G}_{\mathbb{C}}$, relative to a set of intervariable clusters $\mathbb{C}$). See App. A.2 for the formal definitions. Finally, we state the AIC formally below.

**Definition 5** (Abstract Invariance Condition (AIC)). Let $\mathcal{M}_L = \langle \mathbf{U}_L, \mathbf{V}_L, \mathcal{F}_L, P(\mathbf{U}_L) \rangle$ be an SCM and $\tau : \mathcal{D}_{\mathbf{V}_L} \to \mathcal{D}_{\mathbf{V}_H}$ be a constructive abstraction function relative to $\mathbb{C}$ and $\mathbb{D}$. The SCM $\mathcal{M}_L$ is said to satisfy the abstract invariance condition (AIC, for short) with respect to $\tau$ if, for all $\mathbf{v}_1, \mathbf{v}_2 \in \mathcal{D}_{\mathbf{V}_L}$ such that $\tau(\mathbf{v}_1) = \tau(\mathbf{v}_2)$, $\forall \mathbf{u} \in \mathcal{D}_{\mathbf{U}_L}, \mathbf{C}_i \in \mathbb{C}$, the following holds:

$$\tau_{\mathbf{C}_i} \left( \left( f^L_V(\mathbf{pa}^{(1)}_V, \mathbf{u}_V) : V \in \mathbf{C}_i \right) \right) = \tau_{\mathbf{C}_i} \left( \left( f^L_V(\mathbf{pa}^{(2)}_V, \mathbf{u}_V) : V \in \mathbf{C}_i \right) \right), \tag{1}$$

where $\mathbf{pa}^{(1)}_V$ and $\mathbf{pa}^{(2)}_V$ are the values corresponding to $\mathbf{v}_1$ and $\mathbf{v}_2$. ∎

Intuition for the AIC in the context of this paper is provided in Ex. 2.

## 2 INVARIANCES IN CAUSAL ABSTRACTIONS

Causal abstractions are useful since they provide a framework for bridging the gap between models of different granularities, allowing one to work in a simpler high-level space despite having complicated data from the low-level space. The task of performing causal inferences across abstractions is well-studied in the case where the abstraction function $\tau$ is given. When the inter/intravariable clusters $\mathbb{C}$ and $\mathbb{D}$ are provided alongside the structural assumptions of a graphical model $\mathcal{G}_{\mathbb{C}}$, one can straightforwardly construct $\tau$ and then make high-level inferences using low-level data.

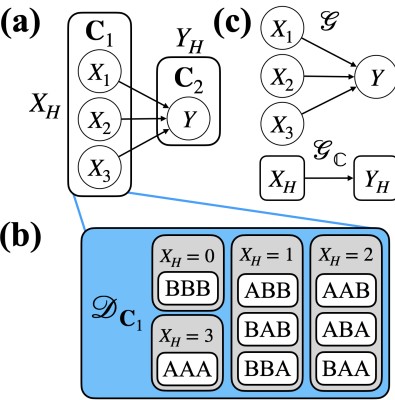

**Example 1.** Suppose a country is voting to elect an official, deciding between candidate A and B. Votes are collected from three districts, $X_1$, $X_2$, and $X_3$, and the outcome of the election $(Y)$ is based on which candidate receives the most votes. On the low level, $\mathbf{V}_L = \{X_1, X_2, X_3, Y\}$, all with a domain of $\{A, B\}$. Instead of collecting data on individual district votes, one may wish to abstract the votes into a single variable representing their sum (i.e., $X_H = \tau(X_1, X_2, X_3) = X_1 + X_2 + X_3$). This corresponds to the intervariable clusters $\mathbb{C} = \{\mathbf{C}_1 = \{X_1, X_2, X_3\}, \mathbf{C}_2 = \{Y\}\}$, shown in Fig. 3(a). The high level variables $X_H$ and $Y_H$ correspond to the clusters $\mathbf{C}_1$ and $\mathbf{C}_2$. The intravariable clusters over $\mathbf{C}_1$ would be $\mathbb{D}_{\mathbf{C}_1} =$

Figure 3: Visualization of Ex. 1. **(a)** On the intervariable level, $X_1$, $X_2$, and $X_3$ are clustered together to form $X_H$, while $Y$ is clustered by itself. **(b)** On the intravariable level, the 8 possible values of $\mathbf{C}_1 = \{X_1, X_2, X_3\}$ are clustered based on the number of votes for $A$. **(c)** The corresponding causal diagram $\mathcal{G}$ and C-DAG $\mathcal{G}_{\mathbb{C}}$.

$\{\{BBB\}, \{ABB, BAB, BBA\}, \{AAB, ABA, BAA\}, \{AAA\}\}$, with the 4 sets corresponding to the values of $X_H = 0$, 1, 2, and 3 respectively (Fig. 3(b)). Then the abstraction is quite natural, with $(X_H, Y_H) \leftarrow \tau(X_1, X_2, X_3, Y) = (X_1 + X_2 + X_3, Y)$. The corresponding causal diagram $\mathcal{G}$ and C-DAG $\mathcal{G}_{\mathbb{C}}$ are shown in Fig. 3(c). ∎

In practice, it may not be the case that $\mathbb{C}$ and $\mathbb{D}$ are readily available. For intervariable clusters $\mathbb{C}$, it is often the case that the clusters are fixed in advance when deciding on the assumptions of the graphical model $\mathcal{G}_{\mathbb{C}}$. The C-DAG $\mathcal{G}_{\mathbb{C}}$ over $\mathbb{C}$ can be much simpler to specify than the full causal diagram $\mathcal{G}$, which requires a full specification of every pairwise relationship in $\mathbf{V}_L$. Given the prevalence of hierarchical structures in data, it can often be quite intuitive which choices of clusters make sense. If all else fails, intervariable clusters can be chosen through a heuristical approach (see (Xia and Bareinboim, 2024, Alg. 3)).

Specifying intravariable clusters $\mathbb{D}$ is a much more difficult challenge. In extremely high-dimensional scenarios such as those involving image data, the size of the domain can become prohibitively large (e.g., a $128 \times 128 \times 3$ image with 256 possible pixel values has $256^{128 \times 128 \times 3}$ different values in its domain). Specifying a partition over such a large space is intractable in general since doing so would require enumerating each possible image and assigning a corresponding cluster label. It would therefore be desirable to use a machine learning approach to learn intravariable clusters from data in a tractable manner.

Learning intravariable clusters is a representation learning task. For each intervariable cluster $\mathbf{C}_i$, the goal is to find which values of $\mathbf{C}_i$ map to which values of $V_{H,i}$ (i.e., learning the mapping $\tau_{\mathbf{C}_i} : \mathcal{D}_{\mathbf{C}_i} \to \mathcal{D}_{V_{H,i}}$). $V_{H,i}$ can then be interpreted as the representation of $\mathbf{C}_i$. Unfortunately, there are strict requirements on what kinds of representations are allowed, shown by the following result.

**Proposition 1** ((Xia and Bareinboim, 2024, Prop. 5)). *Consider a low level SCM $\mathcal{M}_L$ and constructive abstraction function $\tau$ w.r.t. clusters $\mathbb{C}$ and $\mathbb{D}$. $\mathcal{M}_L$ is guaranteed to satisfy the AIC w.r.t. $\tau$ if and only if $\mathbb{D}_{\mathbf{C}_i} = \{\{\mathbf{c}_i\} : \mathbf{c}_i \in \mathcal{D}_{\mathbf{C}_i}\}$ for all $\mathbf{C}_i \in \mathbb{C}$.* ∎

In words, the only choice of intravariable clusters that is guaranteed to satisfy the AIC (Def. 5) is the one where every value in the domain of $\mathbf{C}_i$ is clustered by itself. Any other set of clusters that group two values together may potentially violate the AIC, which is undesirable since it may result in incorrect causal inferences in the high-level model.

For intuition on why this presents a problem, consider the following example.

**Example 2.** Consider an image classification task where $\mathbf{V}_L = \{I_L, Y\}$ for image $I_L$ and label $Y$. For the sake of simplicity, suppose $Y$ is binary, and $I_L$ can only take three possible values: $i_1$, $i_2$, and $i_3$, shown in Fig. 4. Intuitively, it seems that $i_1$ and $i_2$ are the same image but rotated, so it may be tempting to cluster them into the same high-level value (i.e., $\mathbb{D}_I = \{x_1 = \{i_1, i_2\}, x_2 = \{i_3\}\}$). That is, one may wish to construct high-level representation $I_H$ that takes only two possible values, $x_1$ or $x_2$, where $x_1$ refers to both $i_1$ and $i_2$.

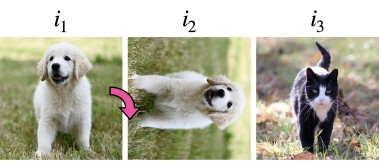

$$i_2 = g_I(i_1, \phi = \pi/2)$$

Figure 4: Three images of $\mathcal{D}_{I_L}$, for Ex. 2. $i_2$ is simply a $\pi/2$ rotation of $i_1$, represented by the invariance function $g_I$.

Unfortunately, without information or assumptions about the underlying causal model, performing this clustering violates the AIC and may result in incorrect inferences. For example, suppose in one possible SCM of the setting, $\mathcal{M}_1$, the function $f_Y^1(i_L, u_Y) = \mathbb{1}\{i_L \in \{i_1, i_2\}\} \oplus u_Y$, while in another, $\mathcal{M}_2$, $f_Y^2(i_L, u_Y) = \mathbb{1}\{i_L \in \{i_1, i_3\}\} \oplus u_Y$. An interpretation might be that in $\mathcal{M}_1$, $Y$ is a label that refers to whether the image is a cat or a dog, while in $\mathcal{M}_2$, $Y$ represents whether the animal in the image is on its side. The proposed clusters for $\mathbb{D}_I$ satisfy the AIC for $\mathcal{M}_1$, but in the case of $\mathcal{M}_2$, clustering these two images leads to ambiguity over whether $x_1$ should receive the label $Y = 0$ or $Y = 1$. However, without additional information about $f_Y$, it is not clear whether $\mathcal{M}_1$ or $\mathcal{M}_2$ (or neither) is the true model. ∎

An implication of Prop. 1 is that the only kinds of representations $\tau_{\mathbf{C}_i}$ that can be learned for each cluster $\mathbf{C}_i$ are ones where $\tau_{\mathbf{C}_i}$ is bijective, also implying that the cardinality of the representation stays the same (i.e., $|\mathcal{D}_{\mathbf{C}_i}| = |\mathcal{D}_{V_{H,i}}|$). Still, this bijectivity requirement is limiting in that it does not allow for dimensionality reduction, one of the main benefits of representation learning.

We now focus on a new strategy of learning intravariable clusters leveraging invariances. Prop. 1 only holds given no additional information about the underlying generating model. However, it may be given that certain invariances hold in the setting. This approach allows for a reduction in the cardinality of the representation without relaxing the AIC definition or removing any causal constraints. We use the concept of cluster coarseness to formalize this idea of dimensionality reduction.

**Definition 6** (Intravariable Cluster Coarsening). Let $\mathbb{D}^1$ and $\mathbb{D}^2$ be two sets of intravariable clusters w.r.t. intervariable clusters $\mathbb{C}$. We say that $\mathbb{D}^2$ is coarser than $\mathbb{D}^1$ (or $\mathbb{D}^1$ is finer than $\mathbb{D}^2$) if for all $\mathbf{C}_i \in \mathbb{C}$ and all $\mathcal{D}_{\mathbf{C}_i}^{j_1} \in \mathbb{D}_{\mathbf{C}_i}^1$, there exists $\mathcal{D}_{\mathbf{C}_i}^{j_2} \in \mathbb{D}_{\mathbf{C}_i}^2$ such that $\mathcal{D}_{\mathbf{C}_i}^{j_1} \subseteq \mathcal{D}_{\mathbf{C}_i}^{j_2}$. ∎

In words, a set of intravariable clusters $\mathbb{D}^2$ is coarser than $\mathbb{D}^1$ if all clusters within $\mathbb{D}^1$ are subsumed by some cluster in $\mathbb{D}^2$. For example, in Ex. 1, one could merge the clusters of $X_H = 2$ and $X_H = 3$ and still conclude that candidate A won from a majority vote. A coarser cluster is therefore more desirable because it implies a lower cardinality in the high-level space. Note that by this definition, all possible sets of intravariable clusters are coarser than the set of individual clusters from Prop. 1. The goal is to see when it is possible to obtain coarser clusters without violating the AIC.

Invariances are used throughout the deep learning literature to improve the efficiency of models for high-dimensional data with rich patterns. For example, in computer vision, many image tasks are assumed to be invariant to rotation, translation, scale, cropping, and jitter (Hadsell et al., 2006; Krizhevsky et al., 2017). In recurrent tasks like with language, it is assumed that a prediction is invariant to all information outside of the context window (Bengio et al., 2000) (temporal invariance). For tasks related to sets and pooling, often permutation invariance can be applied (Zaheer et al., 2017; Murphy et al., 2019). In these tasks, instead of working on the raw data, it is often beneficial to work on a simpler representation that removes unnecessary information by incorporating all of these invariances. We formally define how invariances are interpreted in this work below.

**Definition 7** (Structural Invariance). Given intervariable cluster $\mathbf{C}_i \in \mathbb{C}$ over variables $\mathbf{V}_L$, define $\mathbf{Ch}_{\mathbf{C}_i} = \{V \in \mathbf{V}_L : V \notin \mathbf{C}_i, \mathbf{Pa}_V \cap \mathbf{C}_i \neq \emptyset\}$ as the children of $\mathbf{C}_i$. Let $g_{\mathbf{C}_i} : \mathcal{D}_{\mathbf{C}_i} \times \mathcal{D}_\phi \to \mathcal{D}_{\mathbf{C}_i}$ be a function (with parameters $\phi$) that transforms a value of $\mathbf{C}_i$ to another value of $\mathbf{C}_i$. $g_{\mathbf{C}_i}$ is said to be a structural invariance of SCM $\mathcal{M}_L = \langle \mathbf{U}_L, \mathbf{V}_L, \mathcal{F}_L, P(\mathbf{U}_L) \rangle$ for $\mathbf{C}_i$ iff, for all $V \in \mathbf{Ch}_{\mathbf{C}_i}$, $\phi \in \mathcal{D}_\phi$, $\mathbf{u}_V \in \mathcal{D}_{\mathbf{U}_V}$, $\mathbf{c}_i \in \mathcal{D}_{\mathbf{C}_i}$, and $\mathbf{z} \in \mathcal{D}_{\mathbf{Pa}_V \setminus \mathbf{C}_i}$,

$$f_V^L(\mathbf{c}_i[\mathbf{Pa}_V], \mathbf{z}, \mathbf{u}_V) = f_V^L(g_{\mathbf{C}_i}(\mathbf{c}_i, \phi)[\mathbf{Pa}_V], \mathbf{z}, \mathbf{u}_V). \quad (2)$$

∎

In words, $g_{\mathbf{C}_i}$ is a structural invariance of $\mathcal{M}$ if transforming values of $\mathbf{C}_i$ with $g_{\mathbf{C}_i}$ does not affect the output of the functions of any of its children. Taking advantage of these structural invariances, we define the following set of intravariable clusters which group values based on available invariance information.

**Definition 8** (Maximal Invariance Clusters). Let $\mathbb{I} = \{g_{\mathbf{C}_{i_k}}^k\}_{k=1}^{\ell}$ be a set of structural invariances of SCM $\mathcal{M}$ for some intervariable cluster in $\mathbb{C}$ (each $g^k$ could apply to a different cluster $\mathbf{C}_{i_k}$). For each $\mathbf{C}_i \in \mathbb{C}$, define $\mathbb{D}_{\mathbf{C}_i}^{\mathbb{I}}$ as the partition over $\mathcal{D}_{\mathbf{C}_i}$ relative to the closure of $\mathbb{I}$. That is, for any $\mathcal{D}_{\mathbf{C}_i}^j \in \mathbb{D}_{\mathbf{C}_i}^{\mathbb{I}}$, $\mathbf{c}_a, \mathbf{c}_b$ are both in $\mathcal{D}_{\mathbf{C}_i}^j$ if and only if there exists a sequence $\mathbf{c}_1 = \mathbf{c}_a, \mathbf{c}_2, \mathbf{c}_3, \ldots, \mathbf{c}_N = \mathbf{c}_b$ such that for each $\ell \in \{1, \ldots, N-1\}$, there exists $g_{\mathbf{C}_i}^k$ and some $\phi_k \in \mathcal{D}_{\phi_k}$ such that either $g_{\mathbf{C}_i}^k(\mathbf{c}_\ell, \phi_k) = \mathbf{c}_{\ell+1}$ or $g_{\mathbf{C}_i}^k(\mathbf{c}_{\ell+1}, \phi_k) = \mathbf{c}_\ell$. Then, the intravariable clusters $\mathbb{D} = \{\mathbb{D}_{\mathbf{C}_i}^{\mathbb{I}} : \mathbf{C}_i \in \mathbb{C}\}$ are called the maximal invariance clusters of $\mathbb{I}$. ∎

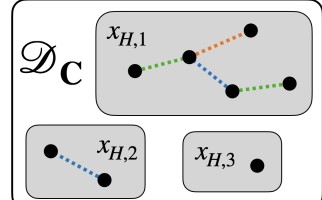

Figure 5: An illustration of constructing the maximal invariance clusters. The values of the intervariable cluster $\mathbf{C}$ (black dots) are connected to each other (via dotted lines) through functions $g^k \in \mathbb{I}$ (each color representing a different $k$). Values that are connected together in some way form an intravariable cluster that defines a high-level value for $X_H = \tau(\mathbf{C})$.

In words, two values are clustered together in the maximal invariance clusters if they are connected through a series of any of the available structural invariances. Intuitively, one can imagine a graph connected by the functions of $\mathbb{I}$, as illustrated in Fig. 5. Values (nodes) are connected with edges corresponding to functions $g^k \in \mathbb{I}$ (e.g., an edge is added between $\mathbf{c}_1$ and $\mathbf{c}_2$ if $\mathbf{c}_1 = g^k(\mathbf{c}_2, \phi_k)$ or $\mathbf{c}_2 = g^k(\mathbf{c}_1, \phi_k)$ for some $g^k$ and $\phi_k$). The corresponding maximal invariance clusters are simply the connected components of the graph.

**Example 3.** Continuing Ex. 1, note that $Y$ is *permutation invariance* to $X_1, X_2, X_3$ (i.e., the order of the votes does not matter). One can define a structural invariance $g_X(X_1, X_2, X_3, \phi)$ where $\phi$ indicates some permutation of the three values. Then, the clusters chosen in Fig. 3 correspond to the maximal invariance clusters of $\mathbb{I} = \{g_X\}$. ∎

It turns out that despite potentially clustering infinite values together, the maximal invariance clusters always satisfy the AIC, as shown next.

**Theorem 1** (Invariance Abstraction Connection). *Let $\mathbb{I}$ be a set of structural invariances of SCM $\mathcal{M}_L$. Then $\mathcal{M}_L$ satisfies the AIC w.r.t. intervariable clusters $\mathbb{C}$ and the maximal invariance clusters $\mathbb{D}$ of $\mathbb{I}$.* ∎

The maximal invariance clusters are maximal in the sense that no coarser cluster is guaranteed to satisfy the AIC with the same set of structural invariances, as shown next.

**Corollary 1.** *$\mathcal{M}_L$ may not satisfy the AIC w.r.t. $\mathbb{C}$ and $\mathbb{D}'$ of structural invariances $\mathbb{I}$ for any $\mathbb{D}'$ that is coarser than the maximal invariance clusters $\mathbb{D}$, and $\mathbb{D}' \neq \mathbb{D}$.* ∎

The concept of maximal invariance clusters is powerful since it provides a much coarser set of clusters that nontrivially reduces the representation size given information about invariances, which is often intuitively assumed to hold in many high-dimensional data settings.

**Example 4.** Continuing Ex. 2, suppose we are given that $f_Y$ is rotationally invariant to the image input $I_L$. This implies that $g_I(i, \phi)$, which rotates $i$ by $\phi$ radians, is a structural invariance of $\mathcal{M}_L$. In this case, the maximal invariance clusters of $\mathbb{I} = \{g_I\}$ is the originally proposed set of clusters $\mathbb{D}_I = \{x_1 = \{i_1, i_2\}, x_2 = \{i_3\}\}$ because $i_2 = g_I(i_1, \phi = \pi/2)$. By Thm. 1, we can therefore eliminate the possibility that $\mathcal{M}_L = \mathcal{M}_2$ and conclude that $\mathbb{D}$ does indeed satisfy the AIC. ∎

Nonetheless, the uniqueness of the maximal invariance clusters makes it difficult to achieve that specific set of clusters in practice. The following two results help relax this requirement.

**Corollary 2.** *$\mathcal{M}_L$ is guaranteed to satisfy the AIC w.r.t. $\mathbb{C}$ and $\mathbb{D}'$ for any $\mathbb{D}'$ that is finer than the maximal invariance clusters $\mathbb{D}$ of structural invariances $\mathbb{I}$.* ∎

**Corollary 3.** *Let $\mathbb{I}_1$ and $\mathbb{I}_2$ be two sets of structural invariances of SCM $\mathcal{M}_L$ such that $\mathbb{I}_1 \subseteq \mathbb{I}_2$ (i.e., there are more invariances in $\mathbb{I}_2$ than $\mathbb{I}_1$). Then, the maximal invariance clusters of $\mathbb{I}_2$ is a coarsening of the maximal invariance clusters of $\mathbb{I}_1$.* ∎

Corol. 2 implies that the AIC is still satisfied even if not all aspects of the invariances are accounted for and a finer set of clusters is learned instead of the maximal one. Corol. 3 implies that the AIC will still hold even if not all of the possible invariances in $\mathbb{I}$ are accounted for. The maximal invariance clusters continue to become increasingly coarse as more invariance functions are added, implying that taking into account more invariances allows for greater dimensionality reduction at no risk of AIC violations.

**Example 5.** Continuing Ex. 1, suppose we are given another structural invariance $g'_X$ such that $g'_X(\text{AAA}) = \text{AAB}$. Incorporating this invariance into the maximal invariance clusters would merge the $X_H = 3$ cluster with the $X_H = 2$ cluster. Note that this is indeed a coarsening of the original clusters, consistent with Corol. 3. Moreover, even though the coarser clusters satisfy the AIC, Corol. 3 guarantees that the original clusters do as well. ∎

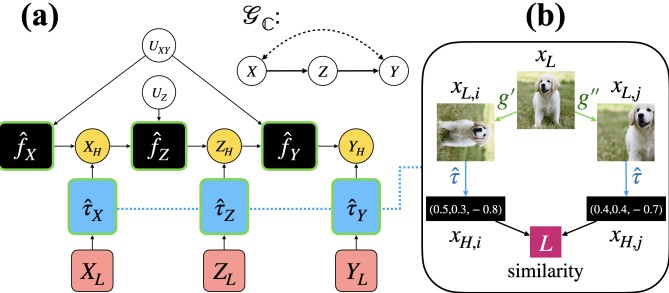

Figure 6: **(a)** An example construction of a $\mathcal{G}_{\mathbb{C}}$-RNCM. Data is given in low-level form ($\mathbf{V}_L$, at bottom in red) and is mapped to high-level form ($\mathbf{V}_H$, in yellow) through neural networks $\hat{\tau}$. Structural functions $\hat{f}$ are neural networks that take inputs according to $\mathcal{G}_{\mathbb{C}}$ and are trained to output their respective variables. **(b)** An example of contrastive learning applied for training $\hat{\tau}$ in an RNCM. A low-level sample $x_L$ is transformed through structural invariances $g'$ and $g''$ to achieve two transformed samples $x_{L,i}$ and $x_{L,j}$. These samples are passed through neural abstraction function $\hat{\tau}$ to produce representations $x_{H,i}$ and $x_{H,j}$, which are compared for similarity in the loss function.

## 3 CONTRASTIVE LEARNING FOR ABSTRACTIONS

Thm. 1 establishes that the maximal invariance clusters obtained through a set of structural invariances will satisfy the AIC. In this section, we explore how to perform representation learning to obtain these clusters in practice. Many sources in the deep learning literature have tackled the interesting but challenging problem of learning invariances, and we leverage the celebrated approach of contrastive learning, following the presentation of Chen et al. (2020).

For causal modeling, we leverage the $\mathcal{G}_{\mathbb{C}}$-constrained representational neural causal model ($\mathcal{G}_{\mathbb{C}}$-RNCM) (Xia and Bareinboim, 2024), which constructs an SCM using neural networks to fit a given C-DAG $\mathcal{G}_{\mathbb{C}}$ (based on intervariable clusters $\mathbb{C}$). An example architecture is shown in Fig. 6(a). Data is provided from the low-level variables $\mathbf{V}_L$, and for each $X_L \in \mathbf{V}_L$, a neural network abstraction function $\hat{\tau}_X$ maps $X_L$ to its high-level representation $X_H \in \mathbf{V}_H$. For each $X_H$, a structural function $\hat{f}_X$ outputs values of $X_H$ according to inputs specified by $\mathcal{G}_{\mathbb{C}}$. Exogenous variables are sampled from a random distribution such as $N(0,1)$ or $\text{Unif}(0,1)$. Collectively, these exogenous variables combined with the structural functions form an SCM that models the high-level variables $\mathbf{V}_H$.

The RNCM follows a two-step training procedure. In the first step, the abstraction functions $\hat{\tau}$ must be trained to learn a representation $X_H$ for each $X_L \in \mathbf{V}_L$. Following the results of Sec. 2, we use contrastive learning in this step to learn invariances for a simpler and more robust representation compared to previous methods of training RNCMs. Fig. 6(b) illustrates this process. Given a low-level sample $x_L \in \mathcal{D}_{X_L}$, $x_L$ is transformed through structural invariances $g', g'' \in \mathbb{I}$ to obtain $x_{L,i}, x_{L,j} \in \mathcal{D}_{X_L}$ ($g'$ and $g''$ can be any composition of functions in $\mathbb{I}$ with any parameters $\phi$). $x_{L,i}$ and $x_{L,j}$ are then mapped through neural network abstraction function $\hat{\tau}_X$ to obtain high-level representation values $x_{H,i}, x_{H,j} \in \mathcal{D}_{X_H}$. Given a batch of $2n$ transformations from $n$ data samples, the following loss function is used.

$$L(x_{H,i}, x_{H,j}) = -\log \frac{\exp\left(\text{sim}(h(x_{H,i}), h(x_{H,j}))/T\right)}{\sum_{k \in \{1,\dots,2n\}: k \neq i} \exp\left(\text{sim}(h(x_{H,i}), h(x_{H,k}))/T\right)}, \tag{3}$$

where $h$ is a neural-parameterized projection head, $\text{sim}$ is any function that computes the similarity of its inputs, and $T$ is a temperature hyperparameter. We leverage cosine similarity for comparing representations, defined as $\text{sim}(z_i, z_j) = \frac{z_i \cdot z_j}{\|z_i\|\|z_j\|}$ for vectors $z_i, z_j$.

An interesting aspect of this loss is that negative samples are not explicitly penalized. Two values that are not intended to be clustered together have representations that are expected to be different due to the nature of how the loss function handles batches. Each sample is implicitly penalized for having too similar of a representation to other samples in the same batch. Nonetheless, in ideal data and computation settings, one can expect this procedure to achieve the maximal invariance clusters, as shown in the next result.

**Theorem 2.** *Under sufficiently large representation size and batch diversity (see Assumption 1 in App. A for details), a set of intravariable clusters $\mathbb{D}$ minimizes loss from Eq. 3 for a given set of structural invariances $\mathbb{I}$ if and only if $\mathbb{D}$ is the maximal invariance clusters of $\mathbb{I}$.* ∎

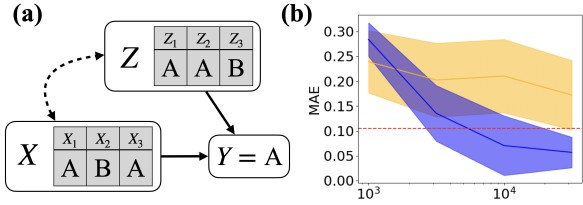

Figure 7: Results for the Votes experiment. **(a)** The C-DAG $\mathcal{G}_{\mathbb{C}}$ for the model. Provinces $X$ and $Z$ each have three districts that vote for their preferred candidate, influencing the outcome of the election $Y$. **(b)** Error at different amounts of data for computing the query $P(Y = \text{A} \mid do(X = (\text{A}, \text{A}, \text{A})))$. The contrastive RNCM (blue, ours) is compared with the original RNCM (orange). The dashed red line shows the error of using the noncausal $P(Y \mid X)$ as the estimate.

In the second step of RNCM training, the structural functions $\hat{f}$ are trained to fit available data on the representation space (e.g., observational data $P(\mathbf{V}_H) = P(\tau(\mathbf{V}_L))$). It is likely that the queries of interest arise from a higher layer of the PCH than the data (e.g., inferring interventional ($\mathcal{L}_2$) quantities from observational ($\mathcal{L}_1$) data). Before inferring these queries, it must be shown that they are identifiable, which can be done through the RNCM model using the NeuralAbstractID algorithm (Xia and Bareinboim, 2024, Alg. 2). Identifiable queries can then be computed directly from the trained RNCM. We leverage the generative adversarial network (GAN) version of the RNCM architecture for training purposes (Xia et al., 2023). We defer the full discussion of RNCM design, training, and inference to prior works, but the details of the models used in this work can be found in App. B.

## 4 EXPERIMENTAL RESULTS

In this section, we validate our findings experimentally. Additional experimental details can be found in App. B. Code will be released after paper acceptance.

### 4.1 VOTING EXPERIMENT

We first test our approach in a synthetic toy experiment. A democratic country is collecting votes to determining who to elect for an office position (C-DAG illustrated in Fig. 7(a)). Votes come from either province $X$ or $Z$, and both provinces have three districts which each have a representative vote. Each vote can go towards candidate A or B, and the outcome ($Y$) will be one of these candidates. The goal is to determine the probability of A winning the election if all votes in $X$ are set to go to A (i.e., $P(Y = \text{A} \mid do(X = (\text{A}, \text{A}, \text{A})))$). Note that there is confounding between the votes of $X$ and $Z$ (a popular candidate will sway the votes of both provinces), so the query is not equivalent to the conditional distribution $P(Y \mid X)$. However, it is identifiable from observational data and the C-DAG (full proof in App. B).

While the values of $X$ and $Z$ are represented by 3-dimensional vectors, we aim to first learn a representation $\tau$ of the two variables and work in the high-level space. The representations take the form of $[0, 1]^2$, so it will be challenging to learn a 2D representation that captures the original 3D inputs. That said, it is noted that the values of $X$ and $Z$ are *permutation invariant*, that is, the order of the values do not matter for deciding $Y$. The contrastive approach is able to leverage a structural invariance $g$ that maps values of $X$ and $Z$ to permutations of itself.

The results are shown in Fig. 7. Our approach (blue) is an RNCM that leverages contrastive learning to learn its embedding, and it is compared to the original RNCM implementation (orange). Note that the contrastive RNCM clearly outperforms the original RNCM, showing significantly lower error with higher samples. In fact, the original RNCM has trouble outperforming the baseline error for incorrectly using $P(Y \mid X)$ as an estimator for $P(Y \mid do(X))$ (dashed red line).

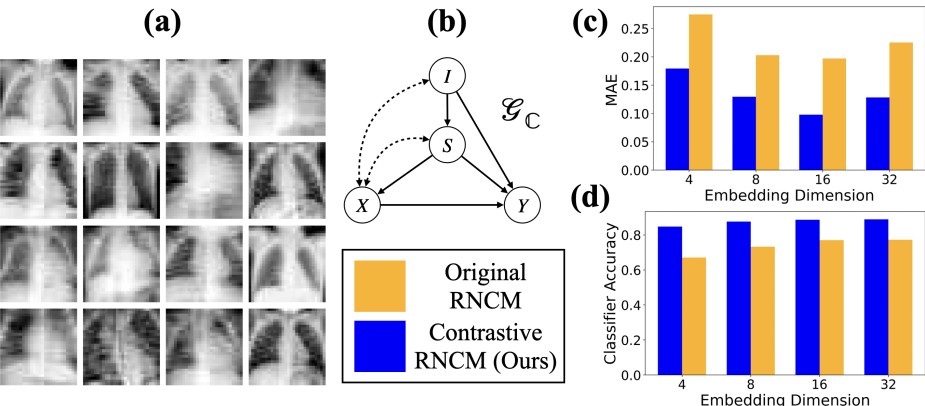

Figure 8: **(a)** Sample x-ray images of $I$. **(b)** C-DAG $\mathcal{G}_{\mathbb{C}}$. **(c)** Comparison of the mean absolute error (MAE) of the query $P(Y \mid do(X), I)$ between the proposed contrastive learning approach (blue) with the original RNCM (orange) across different sizes of embeddings. **(d)** Comparison of the two approaches at classifying $Y$ using $P(Y \mid do(X), I)$ across different sizes of embeddings.

### 4.2 PNEUMONIA EXPERIMENT

We next evaluate our approach on a semi-synthetic medical setting with patient records on chest X-ray images ($I$), pneumonia symptoms ($S$), whether they were given treatment ($X$), and whether they recovered within 30 days ($Y$). The corresponding C-DAG is illustrated in Fig. 8(b). Note that the causal effects from $I$ are not literally from the image pixels themselves but from the underlying conditions captured in the image, and capturing these abstract qualities is one goal of learning the embeddings. Given a chest X-ray image $I = i$, we aim to estimate the causal effect of the treatment $X$, computing the interventional quantity $P(Y = 1 \mid do(x), i)$. Due to unobserved confounding of $X$ with $S$ and $I$, this query differs from the observational $P(Y = 1 \mid x, i)$. Nonetheless, the queries remain identifiable from observational data given the C-DAG (full proof in App. B).

We use approximately 6,000 chest X-ray images of size $28 \times 28$ as provided in (Kermany et al., 2018; Yang et al., 2021; 2023) (examples shown in Fig. 8(a)). We assume that the $I$ is invariant to the transformations presented in (Chen et al., 2020), including translation, zoom, crop, flip, jitter, and blur (i.e., the set of structural invariances $\mathbb{I}$ consist of these transformation functions). Leveraging these invariances, we apply the contrastive learning method from Sec. 3 to learn invariant image embeddings, which are used in the RNCM when fitting the observational data. Using the trained model, we estimate $P(Y = 1 \mid do(x), i)$ and compare with the original RNCM as a baseline.

We vary the dimensionality of the learned embeddings and plot the resulting errors for both approaches. The mean absolute errors (MAE) for both methods are shown in Fig. 8(c). Notably, our approach (blue) significantly outperforms the baseline (orange) across all embedding dimensions, consistently achieving lower MAE. In Fig. 8(d), we also evaluate the quality of the learned embeddings using a simple linear classifier to predict ground truth labels from the original dataset, comparing the accuracies of the two models. With the improved performance of the contrastive RNCM, it is clear that improved embedding quality directly translates to more accurate estimates for high-level causal queries. Interestingly, we note that the classification accuracy of the original RNCM slowly approaches the accuracy of the contrastive RNCM, likely indicating a stronger performance when the embedding size is sufficiently large to avoid AIC violations.

## 5 CONCLUSION

In this paper, we showed how invariance information can allow for lower-dimensional representations in causal abstraction inference (Thm. 1, Corols. 1, 2, 3). We showed how to learn these invariant representations using contrastive learning (Thm. 2), a state-of-the-art tool in noncausal settings. We then demonstrated the strength of these representations empirically, showing how the contrastive RNCM greatly outperforms the original RNCM. This research takes an important step in bridging the gap between state-of-the-art deep learning techniques and causal methods.

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

# A PROOFS

In this section we present the proofs for the technical results of the paper.

## A.1 EXTENDED PRELIMINARIES

Here we provide the full definition of important concepts from the preliminaries section.

**Definition 3** (Inter/Intravariable Clusterings (Xia and Bareinboim, 2024, Def. 5)). Let $\mathcal{M}$ be an SCM over $\mathbf{V}$.

1. A set $\mathbb{C}$ is said to be an intervariable clustering of $\mathbf{V}$ if $\mathbb{C} = \{\mathbf{C}_1, \mathbf{C}_2, \ldots \mathbf{C}_n\}$ is a partition of a subset of $\mathbf{V}$. $\mathbb{C}$ is further considered admissible w.r.t. $\mathcal{M}$ if for any $\mathbf{C}_i \in \mathbb{C}$ and any $V \in \mathbf{C}_i$, no descendent of $V$ outside of $\mathbf{C}_i$ is an ancestor of any variable in $\mathbf{C}_i$. That is, there exists a topological ordering of the clusters of $\mathbb{C}$ relative to the functions of $\mathcal{M}$.

2. A set $\mathbb{D}$ is said to be an intravariable clustering of variables $\mathbf{V}$ w.r.t. $\mathbb{C}$ if $\mathbb{D} = \{\mathbb{D}_{\mathbf{C}_i} : \mathbf{C}_i \in \mathbb{C}\}$, where $\mathbb{D}_{\mathbf{C}_i} = \{\mathcal{D}^1_{\mathbf{C}_i}, \mathcal{D}^2_{\mathbf{C}_i}, \ldots, \mathcal{D}^{m_i}_{\mathbf{C}_i}\}$ is a partition (of size $m_i$) of the domains of the variables in $\mathbf{C}_i$, $\mathcal{D}_{\mathbf{C}_i}$ (recall that $\mathcal{D}_{\mathbf{C}_i}$ is the Cartesian product $\mathcal{D}_{V_1} \times \mathcal{D}_{V_2} \times \cdots \times \mathcal{D}_{V_k}$ for $\mathbf{C}_i = \{V_1, V_2, \ldots, V_k\}$, so elements of $\mathcal{D}^j_{\mathbf{C}_i}$ take the form of tuples of the value settings of $\mathbf{C}_i$). ∎

For clarity, we note that intervariable clusters $\mathbb{C}$ can be a partition of a *subset* of $\mathbf{V}$. That is, variables from $\mathbf{V}$ can be excluded from any cluster in $\mathbb{C}$. In such cases, they are projected away (Lee and Bareinboim, 2019). Additionally, admissibility of $\mathbb{C}$ states that no descendent of $V$ outside of $\mathbf{C}_i$ is an ancestor of any variable in $\mathbf{C}_i$, implying acyclicity among clusters. No statement about descendents inside of $\mathbf{C}_i$ are made.

**Definition 4** (Constructive Abstraction Function (Xia and Bareinboim, 2024, Def. 6)). A function $\tau : \mathcal{D}_{\mathbf{V}_L} \to \mathcal{D}_{\mathbf{V}_H}$ is said to be a constructive abstraction function w.r.t. inter/intravariable clusters $\mathbb{C}$ and $\mathbb{D}$ iff

1. There exists a bijective mapping between $\mathbf{V}_H$ and $\mathbb{C}$ such that each $V_{H,i} \in \mathbf{V}_H$ corresponds to $\mathbf{C}_i \in \mathbb{C}$;

2. For each $V_{H,i} \in \mathbf{V}_H$, there exists a bijective mapping between $\mathcal{D}_{V_{H,i}}$ and $\mathbb{D}_{\mathbf{C}_i}$ such that each $v^j_{H,i} \in \mathcal{D}_{V_{H,i}}$ corresponds to $\mathcal{D}^j_{\mathbf{C}_i} \in \mathbb{D}_{\mathbf{C}_i}$; and

3. $\tau$ is composed of subfunctions $\tau_{\mathbf{C}_i}$ for each $\mathbf{C}_i \in \mathbb{C}$ such that $\mathbf{v}_H = \tau(\mathbf{v}_L) = (\tau_{\mathbf{C}_i}(\mathbf{c}_i) : \mathbf{C}_i \in \mathbb{C})$, where $\tau_{\mathbf{C}_i}(\mathbf{c}_i) = v^j_{H,i}$ if and only if $\mathbf{c}_i \in \mathcal{D}^j_{\mathbf{C}_i}$. We also apply the same notation for any $\mathbf{W}_L \subseteq \mathbf{V}_L$ such that $\mathbf{W}_L$ is a union of clusters in $\mathbb{C}$ (i.e. $\tau(\mathbf{w}_L) = (\tau_{\mathbf{C}_i}(\mathbf{c}_i) : \mathbf{C}_i \in \mathbb{C}, \mathbf{C}_i \subseteq \mathbf{W}_L))$. ∎

## A.2 IMPORTANT DEFINITIONS

Quantities from the distributions of the three layers can be evaluated via the following definitions from Bareinboim et al. (2022).

**Definition 9** (Layer 1 Valuation (Bareinboim et al., 2022, Def. 2)). An SCM $\mathcal{M} = \langle \mathbf{U}, \mathbf{V}, \mathcal{F}, P(\mathbf{U}) \rangle$ defines a joint probability distribution $P^{\mathcal{M}}(\mathbf{V})$ such that for each $\mathbf{Y} \subseteq \mathbf{V}$:

$$P^{\mathcal{M}}(\mathbf{y}) = \int_{\mathcal{D}_{\mathbf{U}}} \mathbb{1}\{\mathbf{Y}(\mathbf{u}) = \mathbf{y}\} dP(\mathbf{u})$$

where $\mathbf{Y}(\mathbf{u})$ is the solution for $\mathbf{Y}$ after evaluating $\mathcal{F}$ with $\mathbf{U} = \mathbf{u}$. ∎

**Definition 10** (Layer 2 Valuation (Bareinboim et al., 2022, Def. 5)). An SCM $\mathcal{M} = \langle \mathbf{U}, \mathbf{V}, \mathcal{F}, P(\mathbf{U}) \rangle$ induces a family of joint distributions over $\mathbf{V}$, one for each intervention $\mathbf{x}$. For each $\mathbf{Y} \subseteq \mathbf{V}$:

$$P^{\mathcal{M}}(\mathbf{y_x}) = \int_{\mathcal{D}_{\mathbf{U}}} \mathbb{1}\{\mathbf{Y_x}(\mathbf{u}) = \mathbf{y}\} dP(\mathbf{u})$$

where $\mathbf{Y_x}(\mathbf{u})$ is the solution for $\mathbf{Y}$ in the submodel $\mathcal{M_x} = \langle \mathbf{U}, \mathbf{V}, \mathcal{F_x}, P(\mathbf{U}) \rangle$, where $\mathcal{F_x} := \{f_V : V \in \mathbf{V} \setminus \mathbf{X}\} \cup \{f_X \leftarrow x : X \in \mathbf{X}\}$. ∎

**Definition 11** (Layer 3 Valuation (Bareinboim et al., 2022, Def. 7)). An SCM $\mathcal{M} = \langle \mathbf{U}, \mathbf{V}, \mathcal{F}, P(\mathbf{U}) \rangle$ induces a family of joint distributions over counterfactual events $\mathbf{Y}_{1[\mathbf{x}_1]}, \mathbf{Y}_{2[\mathbf{x}_2]}, \dots$ for any $\mathbf{Y}_i, \mathbf{X}_i \subseteq \mathbf{V}$:

$$P^{\mathcal{M}}(\mathbf{y}_{1[\mathbf{x}_1]}, \mathbf{y}_{2[\mathbf{x}_2]}, \dots) = \int_{\mathcal{D}_{\mathbf{U}}} \mathbb{1}\{\mathbf{Y}_{1[\mathbf{x}_1]}(\mathbf{u}) = \mathbf{y}_1, \mathbf{Y}_{2[\mathbf{x}_2]}(\mathbf{u}) = \mathbf{y}_2, \dots\} dP(\mathbf{u}).$$

∎

The results of this work are general on all three layers of the PCH.

Every SCM induces a structure called a causal diagram, defined as follows.

**Definition 12** (Causal Diagram (Bareinboim et al., 2022, Def. 13)). Each SCM $\mathcal{M}$ induces a causal diagram $\mathcal{G}$, constructed as follows:

  1. add a vertex for each $V_i \in \mathbf{V}$;

  2. add a directed arrow $(V_j \rightarrow V_i)$ for every $V_i \in \mathbf{V}$ and $V_j \in \mathbf{Pa}_{V_i}$; and

  3. add a dashed-bidirected arrow $(V_j \dashleftarrow\dashrightarrow V_i)$ for every pair $V_i, V_j \in \mathbf{V}$ such that $\mathbf{U}_{V_i}$ and $\mathbf{U}_{V_j}$ are not independent (i.e., unobserved confounding is present). ∎

Given the impossibility of inferring higher layers from lower layers without additional assumptions, many works often assume the availability of the causal diagram and its corresponding implied constraints (possibly in the form of a causal or counterfactual Bayesian network (Bareinboim et al., 2022; Correa and Bareinboim, 2024)). In the context of causal abstractions, a causal diagram on the low-level may be too difficult to specify given the potentially large amount of variables. Instead, a cluster causal diagram is typically assumed instead, defined below.

**Definition 13** (Cluster Causal Diagram (C-DAG) (Anand et al., 2023, Def. 1)). Given a causal diagram $\mathcal{G} = \langle \mathbf{V}, \mathbf{E} \rangle$ and an admissible clustering $\mathbb{C} = \{\mathbf{C}_1, \dots, \mathbf{C}_k\}$ of $\mathbf{V}$, construct a graph $\mathcal{G}_{\mathbb{C}} = \langle \mathbb{C}, \mathbf{E}_{\mathbb{C}} \rangle$ over $\mathbb{C}$ with a set of edges $\mathbf{E}_{\mathbb{C}}$ defined as follows:

  1. A directed edge $\mathbf{C}_i \rightarrow \mathbf{C}_j$ is in $\mathbf{E}_{\mathbb{C}}$ if there exists some $V_i \in \mathbf{C}_i$ and $V_j \in \mathbf{C}_j$ such that $V_i \rightarrow V_j$ is an edge in $\mathbf{E}$.

  2. A dashed bidirected edge $\mathbf{C}_i \leftrightarrow \mathbf{C}_j$ is in $\mathbf{E}_{\mathbb{C}}$ if there exists some $V_i \in \mathbf{C}_i$ and $V_j \in \mathbf{C}_j$ such that $V_i \leftrightarrow V_j$ is an edge in $\mathbf{E}$. ∎

The cluster causal diagram $\mathcal{G}_{\mathbb{C}}$ is constructed relative to a causal diagram $\mathcal{G}$ given intervariable clusters $\mathbb{C}$. It can be thought of as the causal diagram of the high-level model $\mathcal{M}_H$, defined via the constructive abstraction function $\tau$ defined over $\mathbb{C}$.

Quantities between models of different granularities can be compared using the concept of $Q$-$\tau$ consistency, defined below.

**Definition 14.** Denote $\mathbf{Y}_{L,*}$ as a set of counterfactual variables over $\mathbf{V}_L$. That is,

$$\mathbf{Y}_{L,*} = \left( \mathbf{Y}_{L,1[\mathbf{x}_{L,1}]}, \mathbf{Y}_{L,2[\mathbf{x}_{L,2}]}, \dots \right), \tag{4}$$

where each $\mathbf{Y}_{L,i[\mathbf{x}_{L,i}]}$ corresponds to the potential outcomes of the variables $\mathbf{Y}_{L,i}$ under the intervention $\mathbf{X}_{L,i} = \mathbf{x}_{L,i}$. Each $\mathbf{Y}_{L,i}$ and $\mathbf{X}_{L,i}$ must be unions of clusters from $\mathbb{C}$ (i.e. $\mathbf{Y}_{L,i} = \bigcup_{\mathbf{C} \in \mathbb{C}'} \mathbf{C}$ for some $\mathbb{C}' \subseteq \mathbb{C}$) such that $\tau(\mathbf{Y}_{L,i})$ and $\tau(\mathbf{X}_{L,i})$ are well-defined (i.e. $\tau(\mathbf{Y}_{L,i}) = \left( \bigwedge_{\mathbf{C} \in \mathbb{C}'} \tau_{\mathbf{C}}(\mathbf{C}) \right)$). For the high-level counterpart, denote

$$\mathbf{Y}_{H,*} = \tau(\mathbf{Y}_{L,*}) \tag{5}$$

$$= \left( \tau(\mathbf{Y}_{L,1[\tau(\mathbf{x}_{L,1})]}), \tau(\mathbf{Y}_{L,2[\tau(\mathbf{x}_{L,2})]}), \dots \right). \tag{6}$$

For any value $\mathbf{y}_{H,*} \in \mathcal{D}_{\mathbf{Y}_{H,*}}$, denote

$$\mathcal{D}_{\mathbf{Y}_{L,*}}(\mathbf{y}_{H,*}) = \{\mathbf{y}_{L,*} : \mathbf{y}_{L,*} \in \mathcal{D}_{\mathbf{Y}_{L,*}}, \tau(\mathbf{y}_{L,*}) = \mathbf{y}_{H,*}\}, \tag{7}$$

that is, the set of all values $\mathbf{y}_{L,*}$ such that $\tau(\mathbf{y}_{L,*}) = \mathbf{y}_{H,*}$. ∎

**Definition 15** ($Q$-$\tau$ Consistency (Xia and Bareinboim, 2024, Def. 7)). Let $\mathcal{M}_L$ and $\mathcal{M}_H$ be SCMs defined over variables $\mathbf{V}_L$ and $\mathbf{V}_H$, respectively. Let $\tau : \mathcal{D}_{\mathbf{V}_L} \to \mathcal{D}_{\mathbf{V}_H}$ be a constructive abstraction function w.r.t. clusters $\mathbb{C}$ and $\mathbb{D}$. Let

$$Q = \sum_{\mathbf{y}_{L,*} \in \mathcal{D}_{\mathbf{Y}_{L,*}}(\mathbf{y}_{H,*})} P(\mathbf{Y}_{L,*} = \mathbf{y}_{L,*}) \tag{8}$$

be a low-level Layer 3 quantity of interest (for some $\mathbf{y}_{H,*} \in \mathcal{D}_{\mathbf{Y}_{H,*}}$), as expressed in Eq. 4, and let

$$\tau(Q) = P(\mathbf{Y}_{H,*} = \mathbf{y}_{H,*}) \tag{9}$$

be its high level counterpart. We say that $\mathcal{M}_H$ is $Q$-$\tau$ consistent with $\mathcal{M}_L$ if

$$\sum_{\mathbf{y}_{L,*} \in \mathcal{D}_{\mathbf{Y}_{L,*}}(\mathbf{y}_{H,*})} P^{\mathcal{M}_L}(\mathbf{Y}_{L,*} = \mathbf{y}_{L,*})$$
$$= P^{\mathcal{M}_H}(\mathbf{Y}_{H,*} = \mathbf{y}_{H,*}), \tag{10}$$

that is, the value of $Q$ induced by $\mathcal{M}_L$ is equal to the value of $\tau(Q)$ induced by $\mathcal{M}_H$[1]. Furthermore, if $\mathcal{M}_H$ is $Q$-$\tau$ consistent with $\mathcal{M}_L$ for all $Q \in \mathcal{L}_i(\mathcal{M}_L)$ of the form of Eq. 8, then $\mathcal{M}_H$ is said to be $\mathcal{L}_i$-$\tau$ consistent with $\mathcal{M}_L$. ∎

### A.3 PROOFS OF SEC. 2

**Theorem 1** (Invariance Abstraction Connection). *Let $\mathbb{I}$ be a set of structural invariances of SCM $\mathcal{M}_L$. Then $\mathcal{M}_L$ satisfies the AIC w.r.t. intervariable clusters $\mathbb{C}$ and the maximal invariance clusters $\mathbb{D}$ of $\mathbb{I}$.* ∎

*Proof.* Let $\mathbb{I}$ be a set of structural invariances of SCM $\mathcal{M}_L = \langle \mathbf{U}_L, \mathbf{V}_L, \mathcal{F}_L, P(\mathbf{U}_L) \rangle$ with respect to intervariable clusters $\mathbb{C}$. Let $\mathbb{D}$ be the maximal invariance clusters of $\mathbb{I}$. Assume for the sake of contradiction that $\mathcal{M}_L$ does not satisfy the AIC with respect to the constructive abstraction function $\tau$ constructed from $\mathbb{C}$ and $\mathbb{D}$. This implies that for some $\mathbf{C}_i \in \mathbb{C}$, there exists $\mathbf{c}_a, \mathbf{c}_b \in \mathcal{D}_{\mathbf{C}_i}$ such that $\mathbf{c}_a$ and $\mathbf{c}_b$ belong in the same partition in $\mathbb{D}_{\mathbf{C}_i}$, but there is some $f_V^L \in \mathcal{F}_L$ that takes $\mathbf{C}_i$ as input such that

$$\tau_{\mathbf{C}_i}\left( \left( f_V^L(\mathbf{pa}_V^{(a)}, \mathbf{u}_V) : V \in \mathbf{C}_i \right) \right) \neq \tau_{\mathbf{C}_i}\left( \left( f_V^L(\mathbf{pa}_V^{(b)}, \mathbf{u}_V) : V \in \mathbf{C}_i \right) \right), \tag{11}$$

where $\mathbf{pa}_V^{(a)}$ and $\mathbf{pa}_V^{(b)}$ correspond to inputs from $\mathbf{c}_a$ and $\mathbf{c}_b$ respectively.

Suppose that two values, $\mathbf{c}_1, \mathbf{c}_2 \in \mathcal{D}_{\mathbf{C}_i}$ are "linked" if $\mathbf{c}_1 = g^k(\mathbf{c}_2, \phi_k)$ or $\mathbf{c}_2 = g^k(\mathbf{c}_1, \phi_k)$ for some $g^k \in \mathbb{I}$ and $\phi_k \in \mathcal{D}_{\phi_k}$. If it is the latter, then this would imply that for all $\mathbf{u}_V \in \mathcal{D}_{\mathbf{U}_V}$ and $\mathbf{z} \in \mathcal{D}_{\mathbf{Pa}_V \setminus \mathbf{C}_i}$,

$$f_V^L(\mathbf{c}_1[\mathbf{Pa}_V], \mathbf{z}, \mathbf{u}_V) = f_V^L(g^k(\mathbf{c}_1, \phi_k)[\mathbf{Pa}_V], \mathbf{z}, \mathbf{u}_V) = f_V^L(\mathbf{c}_2[\mathbf{Pa}_V], \mathbf{z}, \mathbf{u}_V) \tag{12}$$

$\mathbf{c}_1$ and $\mathbf{c}_2$ can be swapped in the case of the former.

By Def. 8, if $\mathbb{D}$ is the set of maximal invariance clusters of $\mathbb{I}$, then there must exist some sequence of $\mathbf{c}_1, \mathbf{c}_2, \ldots, \mathbf{c}_{\ell-1}$ and $g^1, g^2, \ldots, g^\ell \in \mathbb{I}$ such that $\mathbf{c}_1$ is linked with $\mathbf{c}_a$ through $g^1$, $\mathbf{c}_k$ is linked with $\mathbf{c}_{k-1}$ through $g^k$, and $\mathbf{c}_b$ is linked with $\mathbf{c}_{\ell-1}$ through $g^\ell$.

If $\mathbb{I}$ is a set of structural invariances of $\mathcal{M}_L$, then by definition (Eq. 2), it must be the case that for all $\mathbf{u}_V \in \mathcal{D}_{\mathbf{U}_V}$ and $\mathbf{z} \in \mathcal{D}_{\mathbf{Pa}_V \setminus \mathbf{C}_i}$,

$$f_V^L(\mathbf{c}_a[\mathbf{Pa}_V], \mathbf{z}, \mathbf{u}_V) = f_V^L(\mathbf{c}_1[\mathbf{Pa}_V], \mathbf{z}, \mathbf{u}_V) \tag{13}$$
$$= f_V^L(\mathbf{c}_2[\mathbf{Pa}_V], \mathbf{z}, \mathbf{u}_V) \tag{14}$$
$$= \ldots \tag{15}$$
$$= f_V^L(\mathbf{c}_{\ell-1}[\mathbf{Pa}_V], \mathbf{z}, \mathbf{u}_V) \tag{16}$$
$$= f_V^L(\mathbf{c}_b[\mathbf{Pa}_V], \mathbf{z}, \mathbf{u}_V) \tag{17}$$

following Eq. 12. This contradicts the inequality in Eq. 11 since all such $f_V^L$ must therefore produce the same output for any such $\mathbf{c}_1, \mathbf{c}_2$ in the same cluster. Therefore, the AIC must be satisfied with these clusters. □

---

[1]Note that the equality in Eq. 10 is consistent with the push-forward measure through $\tau$.

**Corollary 1.** $\mathcal{M}_L$ *may not satisfy the AIC w.r.t.* $\mathbb{C}$ *and* $\mathbb{D}'$ *of structural invariances* $\mathbb{I}$ *for any* $\mathbb{D}'$ *that is coarser than the maximal invariance clusters* $\mathbb{D}$, *and* $\mathbb{D}' \neq \mathbb{D}$. ∎

*Proof.* For the premise of this proof, we make no assumptions about the underlying generating model other than that $\mathbb{I}$ is a set of structural invariances of $\mathcal{M}_L$. That is, $\mathcal{M}_L$ can be any SCM such that this applies.

Consider a set of intravariable clusters $\mathbb{D}'$ that is coarser than $\mathbb{D}$ such that $\mathbb{D}' \neq \mathbb{D}$. By Def. 6, this means, for some $\mathbf{C}_i \in \mathbb{C}$, there must exist some $\mathcal{D}^{j_1}_{\mathbf{C}_i}, \mathcal{D}^{j_2}_{\mathbf{C}_i} \in \mathbb{D}_{\mathbf{C}_i}$ and some $\mathcal{D}^{j'}_{\mathbf{C}_i} \in \mathbb{D}'_{\mathbf{C}_i}$ such that $\mathcal{D}^{j_1}_{\mathbf{C}_i} \subset \mathcal{D}^{j'}_{\mathbf{C}_i}$ and $\mathcal{D}^{j_2}_{\mathbf{C}_i} \subset \mathcal{D}^{j'}_{\mathbf{C}_i}$. Consider $\mathbf{c}_1 \in \mathcal{D}^{j_1}_{\mathbf{C}_i}$ and $\mathbf{c}_2 \in \mathcal{D}^{j_2}_{\mathbf{C}_i}$.

Construct $\mathcal{M}_L = \langle \mathbf{U}_L, \mathbf{V}_L, \mathcal{F}_L, P(\mathbf{U}_L) \rangle$ as follows.

1. Define $\mathbf{U}_L$ and $P(\mathbf{U}_L)$ arbitrarily.

2. For some $\mathbf{C}_{i'} \neq \mathbf{C}_i$ and some $X \in \mathbf{C}_{i'}$, define $f^L_X(\mathbf{c}_i) = x_1$ if $\mathbf{c}_i \in \mathcal{D}^{j_1}_{\mathbf{C}_i}$ and $f^L_X(\mathbf{c}_i) = x_2$ for all other $\mathbf{c}_i \in \mathcal{D}_{\mathbf{C}_i}$.

3. For all other $f^L_V$ where $V \in \mathbf{C}_{i'}$, $V \neq X$, define them such that they have no endogenous inputs, and there exists $\mathbf{c}'_1, \mathbf{c}'_2 \in \mathcal{D}_{\mathbf{C}_{i'}}$ such that $\mathbf{c}'_1[X] = x_1$ and $\mathbf{c}'_2[X] = x_2$, and $\mathbf{c}_1$ and $\mathbf{c}_2$ are in separate clusters in $\mathbb{D}_{\mathbf{C}_{i'}}$.

4. Define all other functions in $\mathcal{F}_L$ arbitrarily, but with no endogenous inputs.

Note that this construction of $\mathcal{M}_L$ satisfies the AIC with respect to the constructive abstraction function $\tau$ from $\mathbb{C}$ and $\mathbb{D}$. Eq. 1 is trivially satisfied for any $f^L_V$ where $V \notin \mathbf{C}_i$ since it does not belong in the input set of any other function. For $f^L_X$, note that it will output the same value for any set of inputs $\mathbf{c}_i \in \mathcal{D}_{\mathbf{C}_i}$ that belong in the same cluster, so Eq. 1 is also satisfied for $f^L_V$ where $V \in \mathbf{C}_i$.

However, $\mathcal{M}_L$ clearly does not satisfy the AIC for $\tau$ from $\mathbb{C}$ and $\mathbb{D}'$. Note that $f^L_X(\mathbf{c}_1) = x_1$ and $f^L_V(\mathbf{c}_2) = x_2$, and $\tau(\mathbf{c}'_1) \neq \tau(\mathbf{c}'_2)$. Therefore, it is not guaranteed that any coarser clustering than $\mathbb{D}$ will allow for the AIC to be satisfied. □

For the next proof, first consider the following result.

**Lemma 1** ((Xia and Bareinboim, 2024, Lem. 6)). *For any choice of intravariable clusters* $\mathbb{D}$ *such that* $\mathcal{M}_L$ *satisfies the AIC w.r.t. the corresponding* $\tau$, $\mathcal{M}_L$ *will also satisfy the AIC w.r.t. any finer clustering* $\mathbb{D}'$. ∎

**Corollary 2.** $\mathcal{M}_L$ *is guaranteed to satisfy the AIC w.r.t.* $\mathbb{C}$ *and* $\mathbb{D}'$ *for any* $\mathbb{D}'$ *that is finer than the maximal invariance clusters* $\mathbb{D}$ *of structural invariances* $\mathbb{I}$. ∎

*Proof.* This directly follows from Thm. 1 and Lemma 1. □

**Corollary 3.** *Let* $\mathbb{I}_1$ *and* $\mathbb{I}_2$ *be two sets of structural invariances of SCM* $\mathcal{M}_L$ *such that* $\mathbb{I}_1 \subseteq \mathbb{I}_2$ *(i.e., there are more invariances in* $\mathbb{I}_2$ *than* $\mathbb{I}_1$*). Then, the maximal invariance clusters of* $\mathbb{I}_2$ *is a coarsening of the maximal invariance clusters of* $\mathbb{I}_1$. ∎

*Proof.* Denote $\mathbb{D}_1$ and $\mathbb{D}_2$ as the maximal invariance clusters of $\mathbb{I}_1$ and $\mathbb{I}_2$ respectively. If $\mathbb{I}_1 \subseteq \mathbb{I}_2$, then if two values are in the same cluster in $\mathbb{D}_1$, they must also be in the same cluster in $\mathbb{D}_2$. This is because there must be some sequence of functions in $\mathbb{I}_1$ that link the two values (as in the proof of Thm. 1), and those same functions must exist in $\mathbb{I}_2$.

Trivially, if $\mathbb{I}_2 = \mathbb{I}_1$, then $\mathbb{D}_2 = \mathbb{D}_1$, so it must be a coarsening. Otherwise, starting with the baseline of $\mathbb{D}_1$, consider two values $\mathbf{c}_a, \mathbf{c}_b \in \mathbf{C}_i$ such that they are linked by some function $g^k \in \mathbb{I}_2 \setminus \mathbb{I}_1$, that is, either $\mathbf{c}_a = g^k(\mathbf{c}_b, \phi_k)$ or $\mathbf{c}_b = g^k(\mathbf{c}_a, \phi_k)$ for some $\phi_k \in \mathcal{D}_{\phi_k}$. If $\mathbf{c}_a$ and $\mathbf{c}_b$ are in the same cluster in $\mathbb{D}_1$, then this function is redundant, and nothing is changed in $\mathbb{D}_2$. Otherwise, $\mathbf{c}_a$ and $\mathbf{c}_b$ are linked through $g^k$, implying that all values of $\mathbf{C}_i$ in the same cluster as $\mathbf{c}_a$ can be connected through some sequence of functions in $\mathbb{I}_2$ with all values of in the same cluster as $\mathbf{c}_b$, merging the two clusters together in $\mathbb{D}_2$. Given that any additional function in $\mathbb{I}_2$ can only merge clusters of $\mathbb{D}_1$ into larger and larger clusters, $\mathbb{D}_2$ must be a coarsening of $\mathbb{D}_1$. □

## A.4 PROOFS OF SEC. 3

**Assumption 1.** We assume the following for proofs in this section.

    (a) $\text{sim}(z_i, z_j)$ is maximized if and only if $z_i = z_j$

    (b) $T > 0$

    (c) $h$ is bijective

    (d) Every value of $\mathcal{D}_{\mathbf{V}_L}$ is either in the training dataset or can be achieved through a series of transformations from $\mathbb{I}$ on some point in the dataset.

    (e) [Sufficient Representation Size] For any intervariable cluster $\mathbf{C} \in \mathbb{C}$ and corresponding high-level representation $X_H$, $|\mathcal{D}_{X_H}|$ is sufficiently high-dimensional such that there exists a mapping $\tau_{\mathbf{C}} : \mathcal{D}_{\mathbf{C}} \to \mathcal{D}_{X_H}$ where $\tau(\mathbf{c}_1) = \tau(\mathbf{c}_2)$ only if $\mathbf{c}_1$ and $\mathbf{c}_2$ belong in different intravariable clusters.

    (f) [Sufficient Batch Diversity] For any two original data points $\mathbf{v}_1, \mathbf{v}_2$ provided in the batch and every intervariable cluster $\mathbf{C} \in \mathbb{C}$, the corresponding values $\mathbf{c}_1, \mathbf{c}_2$ belong in different clusters (i.e., two values in the batch are only in the same cluster following the transformation step if one is a transformation of the other).

We argue that these assumptions are reasonable. Assumptions (a) and (b) are in place to avoid unexpected behavior in the mathematical definitions. Assumption (c) is without loss of generality, since technically $h$ can be subsumed into $\tau$ otherwise. Assumption (d) ensures that the lack of convergence is not due to a lack of data. Assumption (e) places a natural limitation on the size of the representation: it must at least be as large as the intended number of clusters. Otherwise, values may be clustered together simply due to the pigeonhole principle. Assumption (f) ensures that, as intended by Eq. 3, two values are expected to be qualitatively different if they are not transformations of each other. In cases with high-dimensional data like with images, this is almost guaranteed to be the case since it is highly unlikely that one image is a transformation of another image in the original dataset. In low-dimensional cases, this assumption may not hold, so we provide Corol. 4 to show how violations of this assumption affect the validity of the clusters. ∎

**Theorem 2.** *Under sufficiently large representation size and batch diversity (see Assumption 1 in App. A for details), a set of intravariable clusters $\mathbb{D}$ minimizes loss from Eq. 3 for a given set of structural invariances $\mathbb{I}$ if and only if $\mathbb{D}$ is the maximal invariance clusters of $\mathbb{I}$.* ∎

*Proof.* For this proof, we make the assumptions in Assumption 1. For short, we denote each assumption as A1(a)-(f).

For simplicity, consider a single intervariable cluster $\mathbf{C} \in \mathbb{C}$, since the loss can be applied independently for each cluster. For this cluster $\mathbf{C}$, denote $x_{H,i}$ and $x_{H,j}$ as the representations of $\mathbf{c}_{L,i}$ and $\mathbf{c}_{L,j}$ respectively (i.e., $\tau(\mathbf{c}_{L,i}) = x_{H,i}$, $\tau(\mathbf{c}_{L,j}) = x_{H,j}$). $\mathbf{c}_{L,i}$ and $\mathbf{c}_{L,j}$ are derived from applying transformations (in $\mathbb{I}$) to some original value $\mathbf{c}_L \in \mathcal{D}_{\mathbf{C}}$, and then their high-level representations $x_{H,i}$ and $x_{H,j}$ are evaluated through Eq. 3.

When $h$ is bijective (A1(c)), we can continue the rest of the proof assuming without loss of generality that the similarity function $\text{sim}$ is applied directly on top of the embeddings $x_{H,i}$ and $x_{H,j}$.

By A1(a), A1(b), and the monotonicity of the $\log$ and $\exp$ function, note that Eq. 3 is minimized when $\text{sim}(x_{H,i}, x_{H,j})$ is maximized and $\text{sim}(x_{H,i}, x_{H,k})$ for $i \neq k$ is minimized.

Note that $\mathbf{c}_{L,i}$ and $\mathbf{c}_{L,j}$ are placed in the same intravariable cluster if $x_{H,i} = x_{H,j}$, and Eq. 3 is only applied when $\mathbf{c}_{L,i}$ and $\mathbf{c}_{L,j}$ are intended to be in the same intravariable cluster in the maximal invariance clusters of $\mathbb{I}$, since both $\mathbf{c}_{L,i}$ and $\mathbf{c}_{L,j}$ are transformations of $\mathbf{c}_L$ by some composition of functions in $\mathbb{I}$.

If $\mathbf{c}_{L,i}$ is in the same cluster as $\mathbf{c}_{L,j}$, and it is not in the same cluster as any $\mathbf{c}_{L,k}$ (A1(f)), then any representation such that $x_{H,i} \neq x_{H,j}$ or $x_{H,i} = x_{H,k}$ can change this (A1(e)) to further optimize Eq. 3, concluding the proof. □

We note that there may be additional practical concerns not addressed by this proof. Notably, optimization procedures are not perfect in practice, and they may not find a more optimal set of representations just because they exist. The stochastic nature of batch optimization may affect this as well. Nonetheless, the effectiveness of contrastive learning is generally well-understood (von Kügelgen et al., 2021; Zimmermann et al., 2021).

Note that the above proof requires that $\mathbf{c}_{L,i}$ is not in the same cluster as any $\mathbf{c}_{L,k}$. In practice, this is likely to be true for high-dimensional data settings such as with images, since it is unlikely that any image (or transformation of one) is going to be identical to another image in the same batch. Nonetheless, this may be a concern in discrete low-dimensional data settings. To understand the limitations of Eq. 3, consider the following results.

**Lemma 2.** *If $x, y, c, d > 0$ and $y \geq dx$, then*

$$\frac{x+c}{y+dc} \geq \frac{x}{y}. \tag{18}$$

*Proof.* Observe that

$$x(y+c) = xy + cy \geq xy + c(dx) = x(y+cd), \tag{19}$$

and $(y+c)$ and $(y+cd)$ can be divided from both sides to achieve the result. $\square$

**Corollary 4.** *Denote $s_j$ as $\exp(\mathrm{sim}(x_{H,i}, x_{H,j})/T)$ and $s_k$ as $\exp(\mathrm{sim}(x_{H,i}, x_{H,k})/T)$. Denote $s^* = \max_{z_i,z_j} \mathrm{sim}(z_i, z_j)$, achieved when $z_i = z_j$. Denote $c = s^* - s_j$, and let $D$ be the indices of $k$ of the batch samples that are in the same cluster as $\mathbf{c}_{L,i}$ (i.e., $x_{H,k} = x_{H,i}$ in the intended clusters). Suppose $\sum_{k \in D} s_k \leq dc$. Then, the maximal invariance clusters minimize Eq. 3 if $\sum_k s_k \geq ds_j$.* ∎

*Proof.* For any particular set of representations, the loss of Eq. 3 can be written as

$$L(x_{H,i}, x_{H,j}) = -\log \frac{s_j}{\sum_k s_k}, \tag{20}$$

which is minimized when $\frac{s_j}{\sum_k s_k}$ is maximized. Forcing $x_{H,i} = x_{H,j}$ would result in the value of $\frac{s^*}{\sum_{k \notin D} s_k + \sum_{k \in D} s^*}$, where $D$ represents the $d$ values that are also forced into the same cluster. Now observe that

$$\frac{s^*}{\sum_{k \notin D} s_k + \sum_{k \in D} s^*} \geq \frac{s^*}{\sum_k s_k + dc} \tag{21}$$

$$\geq \frac{s^* - c}{\sum_k s_k} \qquad \text{from Lem. 2} \tag{22}$$

$$= \frac{s_j}{\sum_k s_k}. \tag{23}$$

Therefore, the new clusters with $x_{H,i} = x_{H,j}$ is more optimal with respect to Eq. 3 than any alternative set of clusters. $\square$

## B  EXPERIMENTAL DETAILS

This section provides detailed information about our experiments. Our models were implemented primarily in PyTorch Paszke et al. (2017), and training was facilitated by PyTorch Lightning Falcon and The PyTorch Lightning team (2019).

The models in this paper are based on neural causal models, specifically $\mathcal{G}$-constrained neural causal models, defined below.

**Definition 16** ($\mathcal{G}$-Constrained Neural Causal Model ($\mathcal{G}$-NCM) (Xia et al., 2021, Def. 7)). *Given a causal diagram $\mathcal{G}$, a $\mathcal{G}$-constrained Neural Causal Model (for short, $\mathcal{G}$-NCM) $\widehat{M}(\boldsymbol{\theta})$ over variables $\mathbf{V}$ with parameters $\boldsymbol{\theta} = \{\theta_{V_i} : V_i \in \mathbf{V}\}$ is an SCM $\langle \widehat{\mathbf{U}}, \mathbf{V}, \widehat{\mathcal{F}}, P(\widehat{\mathbf{U}}) \rangle$ such that*

- $\widehat{\mathbf{U}} = \{\widehat{U}_{\mathbf{C}} : \mathbf{C} \in \mathbb{C}(\mathcal{G})\}$, *where $\mathbb{C}(\mathcal{G})$ is the set of all maximal cliques over bidirected edges of $\mathcal{G}$;*

- $\widehat{\mathcal{F}} = \{\hat{f}_{V_i} : V_i \in \mathbf{V}\}$, where each $\hat{f}_{V_i}$ is a feedforward neural network parameterized by $\theta_{V_i} \in \boldsymbol{\theta}$ mapping values of $\mathbf{U}_{V_i} \cup \mathbf{Pa}_{V_i}$ to values of $V_i$ for $\mathbf{U}_{V_i} = \{\widehat{U}_{\mathbf{C}} : \widehat{U}_{\mathbf{C}} \in \widehat{\mathbf{U}} \text{ s.t. } V_i \in \mathbf{C}\}$ and $\mathbf{Pa}_{V_i} = Pa_{\mathcal{G}}(V_i)$;

- $P(\widehat{\mathbf{U}})$ is defined s.t. $\widehat{U} \sim \mathrm{Unif}(0,1)$ for each $\widehat{U} \in \widehat{\mathbf{U}}$. ∎

A $\mathcal{G}$-NCM is a causal generative model that takes the form of a neurally-parameterized SCM, with functions following the graphical structure of $\mathcal{G}$. In particular, in the context of abstractions, we use the representational version, defined below.

**Definition 17** (Representational NCM (RNCM) (Xia and Bareinboim, 2024, Def. 11)). A representational NCM (RNCM) is a tuple $\langle \widehat{\tau}, \widehat{M} \rangle$, where $\widehat{\tau}(\mathbf{v}_L; \boldsymbol{\theta}_\tau)$ is a function parameterized by $\boldsymbol{\theta}_\tau$ mapping from $\mathbf{V}_L$ to $\mathbf{V}_H$, and $\widehat{M}$ is an NCM defined over $\mathbf{V}_H$. A $\mathcal{G}_{\mathbb{C}}$-constrained RNCM ($\mathcal{G}_{\mathbb{C}}$-RNCM) is an RNCM $\langle \widehat{\tau}, \widehat{M} \rangle$ such that $\widehat{\tau}$ is composed of subfunctions $\widehat{\tau}_{\mathbf{C}_i}$ for each $\mathbf{C}_i \in \mathbb{C}$ (each with its own parameters $\boldsymbol{\theta}_{\tau_{\mathbf{C}_i}}$), and $\widehat{M}$ is a $\mathcal{G}_{\mathbb{C}}$-NCM (Def. 16). ∎

That is, a $\mathcal{G}_{\mathbb{C}}$-RNCM is an NCM constructed over the high-level representation $\mathbf{V}_H$, which is learned through neural parameterized functions $\tau$, as discussed in Sec. 3.

### B.1 VOTING EXPERIMENT

In this section, we discuss the experimental setup of the voting experiment in Sec. 4.1.

#### B.1.1 DATA GENERATION

The SCM $\mathcal{M}^* = \mathcal{M}_L = \langle \mathbf{U}_L, \mathbf{V}_L, \mathcal{F}_L, P(\mathbf{U}_L) \rangle$ that was used to generate the data for the experiment can be described as

$$\mathcal{M}^* = \begin{cases} \mathbf{U}_L & = \{U_{XZ} \in [0,1], U_X \in \{0,1\}^3, U_Z \in \{0,1\}^3, U_Y \in \{0,1\}\} \\ \mathbf{V}_L & = \{X \in \{0,1\}^3, Z \in \{0,1\}^3, Y \in \{0,1\}\} \\ \mathcal{F}_L & = \begin{cases} X \leftarrow U_X \\ Z \leftarrow U_Z \\ Y \leftarrow \mathbb{1}\{\mathrm{sum}(X) + \mathrm{sum}(Z) > 3\} \oplus U_Y \end{cases} \\ P(\mathbf{U}_L) & = \begin{cases} U_{XZ} & \sim \frac{\mathrm{Unif}(0,1)+\mathrm{Unif}(0,1)}{2} \\ U_X, U_Z & \sim \mathrm{Bernoulli}(U_{XZ})^3 \\ U_Y & \sim \mathrm{Bernoulli}(0.1) \end{cases} \end{cases}, \quad (24)$$

that is, the votes of $X$ and $Z$ are sampled independently according to a Bernoulli distribution with a bias determined by $U_{XZ}$. Candidates 0 and 1 correspond to B and A respectively. $Y$ indicates a win for candidate 1 if the collected total votes is larger than 3, with $U_Y$ occasionally flipping the result randomly. The C-DAG $\mathcal{G}_{\mathbb{C}}$ is shown in Fig. 7(a).

The query of interest, $P(Y = 1 \mid do(X = (1,1,1)))$, corresponding to $P(Y = A \mid do(X = (A,A,A)))$, is approximately equal to 0.855, which has a 0.105 error compared to the observational $P(Y \mid X) \approx 0.75$.

#### B.1.2 IDENTIFIABILITY OF THE QUERY

Given observational data from $P(\mathbf{V}_L)$, which can be mapped to $P(\mathbf{V}_H)$ through $\tau$, and the C-DAG $\mathcal{G}_{\mathbb{C}}$ in Fig. 7(a), the query $P(Y \mid do(X))$ can be shown to be identifiable. Specifically, backdoor adjustment on $Z$ can be applied, resulting in $P(y \mid do(x)) = \sum_z P(y \mid x, z)P(z)$.

#### B.1.3 MODEL ARCHITECTURE

Both the original RNCM approach and contrastive RNCM approach follow the definition of Def. 17. For $\widehat{\tau}$, applied to $X$ and $Z$, a multilayer perceptron (MLP) is used with 2 16-dimensional hidden layers, with ReLU activations and a 2-dimensional sigmoid output (i.e., constrained between $[0,1]$). For the original RNCM, which uses an autoencoder structure, an inverse $\widehat{\tau}^{-1}$ is used for each $\widehat{\tau}$, also an MLP with 2 16-dimensional hidden layers, ReLU activations, and a 3-dimensional sigmoid output.

For the NCM body, neural networks $\hat{f}_X$, $\hat{f}_Z$, and $\hat{f}_Y$ are constructed to generate $X$, $Z$, and $Y$ respectively, following the graph $\mathcal{G}_\mathbb{C}$. $\hat{f}_X$ and $\hat{f}_Z$ share a 12-dimensional exogenous input $U_{XZ}$ sampled from $\mathrm{Unif}(0,1)^{12}$, and $\hat{f}_Y$ takes $X$, $Z$, and $U_Y \sim \mathrm{Unif}(0,1)^2$. All three neural networks are MLPs with 2 16-dimensional hidden layers, ReLU activations, and sigmoid outputs. $\hat{f}_X$ and $\hat{f}_Z$ are modeled to output the representations $\hat{\tau}(X)$ and $\hat{\tau}(Z)$, which take the form of $[0,1]^2$ and are not rounded at inference time. $Y$ is not mapped through a representation $\hat{\tau}$, so $\hat{f}_Y$ directly outputs samples of $Y$, where the sigmoid outputs are rounded at inference time.

In training, NCMs are implemented using a generative adversarial approach (Goodfellow et al., 2014). During the distribution-learning phase, the NCM serves as the generator, while a separate discriminator (or critic) network is used to compare fake generated samples with the real samples from the data. In this experiment, the discriminator is an MLP with 2 32-dimensional hidden layers, ReLU activations, and real-valued outputs, which takes the entirety of $\mathbf{V}_H$ as input.

In all MLPs, we apply layer normalization after each hidden layer (Ba et al., 2016). All weights are initialized via Glorot initialization (Glorot and Bengio, 2010). Hyperparameters are largely chosen based on recommendations from prior works, but similar hyperparameters flexibly provided similar quality results.

### B.1.4 EXPERIMENTAL PROCEDURE

In the experiment procedure, first, low-level data is generated from the data-generating model from Sec. B.1.1. The model is then instantiated according to Sec. B.1.3. A two part training phase is used, as described in Sec. 3.

In the first phase, the representation networks $\hat{\tau}$ are trained. In each epoch, the dataset is passed in batches of 256 through a forward pass through the $\hat{\tau}$ functions to obtain the representations $X_H$ and $Z_H$. For the contrastive RNCM, the loss in Eq. 3 is computed for the representations (the projection head $h$ is not used for this experiment). In this case, the set of structural invariances $\mathbb{I}$ contains a single function $g$ in which $g(\mathbf{x}_L)$ outputs a permutation of $\mathbf{x}_L$. For the original RNCM, a reconstruction loss is applied leveraging $\hat{\tau}^{-1}$. That is,

$$L(X_L) = d(\hat{\tau}^{-1}(\hat{\tau}(X_L)), X_L), \tag{25}$$

where $d$ is a distance metric (MSE is used in this work). The loss is then backpropagated, and the weights are updated using the Adam optimizer (Kingma and Ba, 2015). A learning rate of $10^{-4}$ was used, and the procedure is run for 200 epochs. A temperature value of $T = 0.01$ is used for the contrastive RNCM.

In the second phase, the NCM is trained to fit the high-level observational data $P(\mathbf{V}_H)$. In each epoch, a fake and a real batch of 128 samples are generated. The real batch is sampled from the data, while the fake batch is generated from the NCM through a forward pass of the NCM functions. Both batches are passed through the discriminator, and both the NCM and the discriminator are then trained using the Wasserstein GAN loss (Arjovsky et al., 2017). A learning rate of $10^{-4}$ is used for the NCM, while $2 \times 10^{-4}$ is used for the discriminator. The procedure is run for 200 epochs.

After models are trained, they are evaluated on the query $P(Y = 1 \mid do(X = (1,1,1)))$, corresponding to the query $P(Y = A \mid do(X = (A, A, A)))$. The NCM is evaluated on $10^5$ Monte-Carlo samples of the query, sampled via Def. 2. The ground truth is sampled similarly but from the data-generating model.

We reran this procedure for different sample sizes $n \in \{10^3, 10^{3.5}, 10^4, 10^{4.5}\}$ and reran each setting 10 times, displaying 95% confidence intervals for the 10 trials. The results are shown in Fig. 7.

The trials of this experiment were run on Nvidia H100 GPUs, requiring approximately 100 GPU hours.

### B.2 PNEUMONIA EXPERIMENT

In this section, we discuss the experimental setup of the pneumonia experiment in Sec. 4.2.

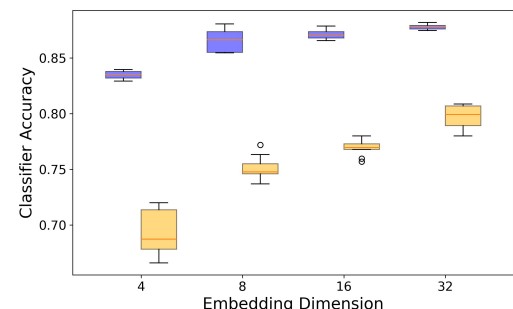 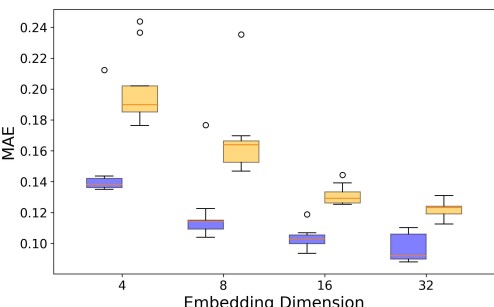

Figure 9: **(a)** Classification accuracy of a linear model trained to predict image labels, using either the contrastive-learning embeddings (blue) or the autoencoder embeddings (orange). **(b)** Mean absolute error (MAE) of the interventional query $P(Y \mid \mathrm{do}(X), I)$ for the proposed contrastive learning approach (blue) versus the original RNCM (orange), evaluated across different embedding dimensionalities. All results are based on $N = 10$ experimental runs and are summarized as box plots.

### B.2.1 Additional Results

To corroborate the findings presented in Fig. 8, we repeated each experiment $N = 10$ times and summarize the outcomes as box plots in Fig. 9.

To evaluate the quality of the learned encoders, we extracted embeddings for every image in the PneumoniaMNIST dataset Kermany et al. (2018); Yang et al. (2021; 2023) and fit a linear classifier to predict the presence of pneumonia using the dataset's ground-truth labels. The resulting accuracies are shown in Fig. 9(a).

### B.2.2 Data Generation

For our pneumonia experiment, we generate synthetic training data from an SCM $\mathcal{M}^* = \mathcal{M}_L = \langle \mathbf{U}_L, \mathbf{V}_L, \mathcal{F}_L, P(\mathbf{U}_L) \rangle$ which can be described as follows

$$\mathcal{M}^* = \begin{cases} \mathbf{V}_L = \{I, S, X, Y\} \\ \mathbf{U}_L = \{U_{IS}, U_{SX}, U_Y\} \\ \mathcal{F}_L = \begin{cases} I \leftarrow h(U_{IS}) \\ S \leftarrow c(I) \wedge ((U_{SX} < 0.75) \vee (U_{SX} > 0.90)) \\ X \leftarrow (S \wedge ((U_1 < 0.25) \vee (U_1 > 0.75))) \\ \quad \vee (\neg S \wedge ((U_2 < 0.35) \oplus (U_1 > 0.45))) \\ Y \leftarrow \begin{bmatrix} \neg (c(I) = 1 \wedge S = 1 \wedge X = 0) \\ \wedge \neg(c(I) = 1 \wedge S = 0 \wedge X = 0) \\ \wedge \neg(c(I) = 0 \wedge S = 0 \wedge X = 1) \\ \wedge \neg(c(I) = 0 \wedge S = 1 \wedge X = 1) \end{bmatrix} \oplus (U_Y < 0.2) \end{cases} \\ P(\mathbf{U}_L) = \{U_{IS}, U_{SX}, U_Y \sim \mathcal{U}[0, 1] \end{cases}, \quad (26)$$

Here, $\oplus$ denotes the logical XOR operator. The function $c(I)$ returns the binary class label corresponding to the presence of pneumonia in the image $I$, i.e., $c(I) \in \{0, 1\}$. The function $h(U_{IS})$ randomly selects an image from class 0 if $U_{IS} < 0.5$, and from class 1 otherwise.

Therefore, each data point corresponds to a patient associated with an X-ray image of their lungs $I$. Based on this image, a binary symptom variable $S$ is inferred, indicating whether the patient exhibits pneumonia symptoms. Depending on the presence or absence of symptoms, the patient might receive treatment $X$. There is unobserved confounding between $I$ and $X$, as well as between $S$ and $X$. Finally, a binary outcome variable $Y$ indicates whether the patient recovered within a month, and it is a function of $I$, $S$, and $X$.

To generate syntetic data from $\mathcal{M}^*$, we use the Pneumonia-MNIST dataset introduced in Kermany et al. (2018); Yang et al. (2021; 2023), which provides X-ray images and corresponding binary labels.

These images serve as a base for generating synthetic data using $\mathcal{M}^*$. The corresponding C-DAG $\mathcal{G}_{\mathbb{C}}$ is shown in Fig. 8(b).

### B.2.3 MODEL ARCHITECTURE

Both the original RNCM and the contrastive RNCM approach follow the structure defined in Def. 17. In both cases, an abstraction function $\widehat{\tau}$ is learned to map the low-level image variable $I$ to its high-level representation $E = \widehat{\tau}_I(I)$.

For the original RNCM approach, the abstraction function $\widehat{\tau}$ is learned jointly with its inverse $\widehat{\tau}^{-1}$ using an autoencoder. The encoder consists of two convolutional layers with 64 and 128 channels, respectively, each followed by a ReLU activation and max-pooling. The resulting feature map is flattened and passed through two fully connected layers to produce the final embedding. The decoder reverses this process, starting with two fully connected layers to reshape the embedding, followed by two transposed convolutional layers that reconstruct the input image. During training we minimize the mean squared error between the input and its reconstruction.

In the contrastive RNCM approach, $\widehat{\tau}$ is learned using the unsupervised contrastive learning objective from Eq. 3. Each image is augmented twice using random resized cropping and discrete rotations, with the resulting views forming a positive pair. The encoder consists of three convolutional layers with increasing channel widths (64, 128, 256), each followed by a ReLU activation and max-pooling. After the convolutional blocks, a dense layer converts the pooled feature maps into a fixed-size vector. This vector is then passed through a projection head consisting of two sequential dense layers with a ReLU activation between them to produce the contrastive embedding. Finally, we $\ell_2$-normalize these embeddings before computing the contrastive loss.

To train the structural functions $\widehat{\mathcal{F}}$ in the GAN-RNCM, we adopt an adversarial training setup in which the generator represents the structural functions of the causal model, and a discriminator (critic) distinguishes real from generated samples Goodfellow et al. (2014). Each function in $\widehat{\mathcal{F}}$ is modeled as a fully connected MLP with ReLU activations and a hidden dimension of 128. The generator is composed of five separate networks, namely $\widehat{f}_E$, $\widehat{f}_S$, $\widehat{f}_X$, $\widehat{f}_Y^{\mathrm{emb}}$, and $\widehat{f}_Y$.

$\widehat{f}_E$ maps a 2-dimensional noise vector $U_1$ to logits over discrete indices into a learned table of image embeddings, using Gumbel-softmax sampling with a temperature of $\tau = 0.5$ to enable differentiable index selection. Rather than generating embeddings directly, $\widehat{f}_E$ produces indices, a design choice we justify in the following paragraphs. $\widehat{f}_S$ takes the selected image embedding and a second 2-dimensional noise vector $U_2$ as input, and is implemented as a 3-layer MLP. $\widehat{f}_X$ receives $U_1$, $U_2$, and $S$, and is modeled as a 2-layer MLP. The image embedding is projected into a lower-dimensional space using $\widehat{f}_Y^{\mathrm{emb}}$, a 4-layer MLP that outputs a 4-dimensional representation. $\widehat{f}_Y$ takes the projected embedding, $X$ and $S$, and an additional noise vector $U_Y$ as input, and is implemented as a 3-layer MLP. The discriminator is a fully connected MLP with two hidden layers of width 128, using ReLU activations. Spectral normalization (Miyato et al., 2018) is applied to each linear layer.

As described in Sec. B.2.4, our experiment involves performing interventions on real images from the dataset. Consider, for example, a query $Q$ that requires intervening on a specific image $I_0$. Given that $Q$ is admissible, we aim to estimate it using the trained GAN-RNCM model. This is achieved through the mutilation procedure described next.

The standard inference process in GAN-RNCM involves sampling the noise variables $U_1$, $U_2$, and $U_Y$, and then generating all variables in the SCM using the learned structural functions. However, to model an intervention on the image variable, we override the output of the image generator $\hat{f}_E$ with $\widehat{\tau}_I(I_0)$. This ensures that all downstream components of the GAN-RNCM operate on the specific intervention-defined embedding. This procedure can be extended to more variables as needed, depending on the structure of the query $Q$. For further details on the mutilation approach, we refer the reader to prior work (Xia et al., 2023; Xia and Bareinboim, 2024).

In practice, however, we observed that this form of intervention introduces distribution shift. Specifically, the embeddings produced by the generator during regular training differ significantly from those injected during mutilation, which are derived from real images. This discrepancy negatively affects the reliability of downstream functions such as $\hat{f}_S$ and $\hat{f}_Y$ when used on out-of-distribution inputs.

To address this, we avoid training the generator to produce image embeddings directly. Instead, we associate each training image with a unique index. During training, the generator is modified to produce such indices instead of actual image embeddings. When downstream functions (e.g., $\hat{f}_S$, $\hat{f}_Y$) require the image embedding, we retrieve the embedding corresponding to the generated index. This ensures that all image embeddings passed to the structural functions during both training and inference correspond to real images, thereby eliminating the distribution shift described previously.

Although this approach restricts the generator from producing entirely new image embeddings, this limitation is acceptable for our experimental setup since we are only interested in evaluating interventional queries that intervene on image embeddings.

### B.2.4 EXPERIMENTAL PROCEDURE

To evaluate the performance of the GAN-RNCM pipeline, we conduct experiments on the PneumoniaMNIST dataset Kermany et al. (2018); Yang et al. (2021; 2023). The dataset is originally imbalanced, with 1,214 images in the minority class and 3,484 in the majority class. To construct a balanced dataset, we randomly subsample 1,214 images from the majority class, resulting in a total of 2,428 images with equal class representation.

Using these images and their associated class labels, we generate synthetic training data following the procedure in Sec. B.2.2. From the resulting dataset, we set aside 228 examples (approximately 10%) as a test set. The test set is balanced across class labels, with 50% positive and 50% negative pneumonia cases. The remaining 2,200 examples are used for training.

As described in Sec. B.2.3, we train two variants of the RNCM model, one using representations learned via unsupervised contrastive learning, and the other using representations from an autoencoder baseline. For each representation type, we train models with embedding dimensionalities of 4, 8, 16, and 32. In both cases, the encoder is trained for 25 epochs using the Adam optimizer Kingma and Ba (2015) with a learning rate of $3 \times 10^{-4}$ and a batch size of 32. For contrastive learning, we use the loss from Eq. 3 with a temperature parameter of $T = 0.1$. The autoencoder baseline is trained using a mean squared reconstruction loss.

In the second phase of training, the GAN-RNCM is optimized to approximate the high-level observational distribution $P(\mathbf{V}_H)$. During each epoch, two batches of data are prepared. The real batch is sampled directly from the training data, and the generated batch is created by sampling noise variables and passing them through the generator, which consists of the structural functions $\widehat{\mathcal{F}}$.

Both the real and generated batches contain 1,024 samples and are passed to the discriminator. The discriminator is trained to assign higher values to real samples and lower values to generated samples. At the same time, the generator is trained to produce samples that are indistinguishable from real data based on the discriminator's output. This procedure follows the WGAN-GP framework (Gulrajani et al., 2017), which regularizes the discriminator through a soft penalty on the gradient norm to enforce a relaxed Lipschitz condition.

The training alternates between updating the generator and the discriminator. For each generator update, the discriminator is updated twice. Optimization is performed using the Adam optimizer (Kingma and Ba, 2015). The learning rate for the generator is set to $2 \times 10^{-5}$, and the learning rate for the discriminator is set to $1 \times 10^{-5}$. This training procedure is repeated for a total of 5,000 epochs. All parameters of the generator and discriminator are updated jointly throughout this phase.

In practice, we observe that training the GAN-RNCMs benefit from incorporating a supervised loss signal with the original adverserial loss. Specifically, at the beginning of each epoch, we perform a supervised update for the structural functions $\hat{f}_S$, $\hat{f}_X$, and $\hat{f}_Y$ using real data from that epoch. Let $E_r$, $S_r$, $X_r$, and $Y_r$ denote the real values of the variables $E$, $S$, $X$, and $Y$, respectively. We then minimize the following supervised losses:

$$\mathcal{L}_{\text{sup}}^S = \mathbb{E}_{E_r, S_r, U_2} \left[ \text{CE} \left( \hat{f}_S(E_r, U_2), S_r \right) \right], \tag{27}$$

$$\mathcal{L}_{\text{sup}}^X = \mathbb{E}_{S_r, X_r, U_1, U_2} \left[ \text{CE} \left( \hat{f}_X(S_r, U_1, U_2), X_r \right) \right], \tag{28}$$

$$\mathcal{L}_{\text{sup}}^Y = \mathbb{E}_{E_r, S_r, X_r, Y_r, U_Y} \left[ \text{CE} \left( \hat{f}_Y(E_r, S_r, X_r, U_Y), Y_r \right) \right], \tag{29}$$

where CE denotes the cross-entropy loss and $U_1$, $U_2$, and $U_Y$ are the i.i.d. noise variables from the definition of the SCM $\mathcal{M}^*$. We optimize these supervised losses using the Adam optimizer Kingma and Ba (2015) with a fixed learning rate of $10^{-3}$.

To evaluate each trained model, we estimate the interventional query $P(Y = 1 \mid I = I_0, \text{do}(X = x))$ for every image $I_0$ in the test set and for both values $x \in \{0, 1\}$. Each estimate is computed using $10^4$ Monte Carlo samples from the trained model. As we will show, this query is identifiable and has a high-level counterpart $P(Y = 1 \mid E = \tau_I(I_0), \text{do}(X = x))$, which can be estimated directly using the learned generative model.

The identifiability follows from an application of Rule 2 of the do-calculus Pearl (2000):

$$P(Y = 1 \mid I = I_0, \text{do}(X = x)) = \sum_s P(Y = 1 \mid I = I_0, \text{do}(X = x), S = s) \cdot P(S = s \mid I_0)$$

$$= \sum_s P(Y = 1 \mid I = I_0, X = x, S = s) \cdot P(S = s \mid I_0).$$

Now, one could further apply Rule 2 to obtain:

$$P(S = s \mid I = I_0) = P(S = s \mid \text{do}(I = I_0)),$$

$$P(Y = 1 \mid I = I_0, \text{do}(X = x), S = s) = P(Y = 1 \mid \text{do}(I = I_0), \text{do}(X = x), \text{do}(S = s)),$$

which can both be estimated using the mutilation procedure described in Section B.2.3. However, following the analysis in (Xia et al., 2023, Appendix B.2), we find that estimating the nested counterfactual $P(Y = 1 \mid \text{do}(I = I_0), \text{do}(X = x))$ directly tends to yield lower error, likely due to avoiding the accumulation of error across multiple estimates.

The quality of each model is assessed by computing the mean absolute error between the estimated and ground truth interventional probabilities, averaged over all test samples. Each configuration is evaluated over 10 independent runs, and results are shown in Fig. 9.

## C  ADDITIONAL EXAMPLES

This section contains additional examples that supplement the main body.

### C.1  EXAMPLES FOR SEC. 2

Table 1 shows examples of structural invariances (Def. 7) for different tasks.

Consider the following example for a more nuanced understanding of maximal invariance clusters relative to a given set of structural invariances.

**Example 6.** Suppose in a company, there are four employees $(X_1, X_2, X_3, X_4)$ who are each trying to decide if they wish to work on project $A$ or $B$ (i.e., $X_1, X_2, X_3, X_4 \in \{A, B\}$). Suppose we would like to cluster the decision of the four employees into a single variable $X_H$, and now the goal is to learn an intravariable clustering of the 16 possible values of the joint tuple $(X_1, X_2, X_3, X_4)$. These variables impact the eventual project direction of the company ($Y \in \{A, B\}$).

To proceed, we must ensure that any two values that are clustered together would not be ambiguous for deciding $Y$ (violating the AIC). Suppose we are given the information that $X_i$ is a higher-ranked employee than $X_j$ for $i > j$, and a higher-ranked employee overwrites the decision of a lower-ranked employee. This can be represented by the structural invariance

$$g((X_1, X_2, X_3, X_4), \phi) = \begin{cases} (X_2, X_2, X_3, X_4) & \phi = 2 \\ (X_1, X_3, X_3, X_4) & \phi = 3 \\ (X_1, X_2, X_4, X_4) & \phi = 4 \end{cases}, \tag{30}$$

| Name | Function Description | Illustration |
|------|---------------------|--------------|
| Permutation Invariance | $g(x, \phi)$ is a reordering of the dimensions of $x$ specified by indices in $\phi$ |  |
| Temporal Invariance | $g(x_t, \phi) = x_{t+\phi}$ for time step $t$ |  |
| Rotational Invariance | $g(i, \phi)$ rotates image $i$ by $\phi$ radians |  |
| Scale Invariance | $g(i, \phi_1, \phi_2)$ zooms image $i$ by $\phi_1$ amount and crops it to region $\phi_2$ |  |
| Translational Invariance | $g(i, \phi)$ pans image $i$ by $\phi$ pixels |  |

Table 1: Examples of invariances and their corresponding structural invariance functions. Many invariances are specifically applicable to the image setting, such as the bottom three on this table.

where $\phi \in \{2, 3, 4\}$ represents an index of $X$. For example, $g((A, B, A, B), 2) = (B, B, A, B)$, indicating that $X_1$ will take the value of $X_2 = B$ even if $X_1$ was originally $A$.

Suppose $\mathbb{I} = \{g\}$ and $\mathbb{D}$ is the maximal invariance clusters of $\mathbb{I}$. Under the definition of maximal invariance clusters, it is therefore the case that $(A, B, A, B)$ and $(B, B, A, B)$ are in the same cluster of $\mathbb{D}$. However, note that $g$ is not reversible in this case (i.e., there is no $\phi$ such that $g((B, B, A, B), \phi) = (A, B, A, B)$.

Interestingly, note that $g((B, B, A, B), 4) = (B, B, B, B)$, putting $(B, B, A, B)$ and $(B, B, B, B)$ in the same cluster in $\mathbb{D}$ as well. This implies that $(A, B, A, B)$ and $(B, B, B, B)$ are in the same cluster despite the lack of direct connection through $g$ in either direction (i.e., there is no $\phi$ such that $g((A, B, A, B), \phi) = (B, B, B, B)$ or $g((B, B, B, B), \phi) = (A, B, A, B)$). Hence, to fully evaluate whether two values are in the same cluster, it must be checked whether there is a path that connects the two values through some series of applications of functions in $\mathbb{I}$, in either direction. ∎

# D  DISCUSSION

This section includes additional discussion points for this work.

## D.1  LIMITATIONS

The results in this work, both theoretical and empirical, are limited by the validity of the assumptions.

Naturally, the most prominent assumption in this paper is the availability of invariance information, with the properties described in Def. 7. Without this information or any other types of assumptions, no set of intravariable clusters can be learned without potentially violating the AIC, as described by Prop. 1. Furthermore, it is possible that the set of available structural invariances, $\mathbb{I}$, does not contain that much helpful information. If the functions are not flexible in terms of mapping to different values given the parameterization $\phi$, it is possible that the corresponding maximal invariance clusters are

still quite fine. Nonetheless, this is the crucial assumption that allows the applicability of the methods of this paper. If this assumption cannot be met, then it is recommended to find alternative solutions to navigate the AIC. Still, this assumption is quite reasonable in any setting in which invariances are naturally assumed to hold anyways, such as rotational invariance in image settings.

In the context of causal abstraction inference, identification of causal queries is crucial for guaranteeing that the causal queries can be inferred from the available information. Notably, the assumption of a graphical model such as the C-DAG $\mathcal{G}_{\mathbb{C}}$ is necessary to avoid issues regarding the Causal Hierarchy Theorem (Bareinboim et al., 2022). Without graphical assumptions (or sometimes even with graphical assumptions), non-identifiability of the desired query would pose a significant issue. Alternative solutions are possible, such as using weaker assumptions for structural learning, or bounding the query rather than precise identification. Still, it is generally the case that the set of inferrable results grows in proportion to the strength of the assumptions.

For contrastive learning, notably Thm. 2, proper representation learning requires a diverse batch such that equivalent values are always compared similarly and different values are always contrasted apart. That is, in the ideal case, any pair of values intended to be in the same cluster will eventually be compared as $x_{H,i}$ and $x_{H,j}$ in Eq. 3, while all other values in the batch are intended to be in different clusters. It is possible that this ideal case is violated, but the maximal invariance clusters are still achieved, as shown in Corol. 4. In higher-dimensional cases like with image data, it is more likely that this is not an issue, since it is unlikely that two different samples in the same batch belong in the same cluster, and a representative set of samples from the invariance functions will eventually be achieved with sufficient training.

Finally, in the context of empirical training, it is always a possibility that training may have issues converging, either due to low compute, underparameterization, or difficulties with gradient-based optimization. This can occur both in the representation training phase and in the generative modeling phase. Failures in the representation training phase are more forgiving, since with a sufficiently large representation dimensionality, this would simply mean a finer set of clusters, which while not ideal, would not violate the AIC. Failures in the generative modeling phase may result in incorrect inferences, but the inferences are guaranteed given proper fitting of the available data, so it is crucial in this phase to ensure that the given data distribution is fitted properly.

### D.2 MOST OPTIMAL CLUSTERS

We note that the maximal invariance clusters given the domains of $\mathcal{D}_{\mathbf{V}_L}$ and a set of structural invariances $\mathbb{I}$ is unique, shown straightforwardly in the following result.

**Proposition 2** (Maximal Invariance Clusters Uniqueness). *The maximal invariance clusters $\mathbb{D}$ of a set of structural invariance $\mathbb{I}$ are unique.*

*Proof.* Assume for the sake of contradiction that $\mathbb{D}$ and $\mathbb{D}'$ are two different sets of intravariable clusters that are both maximal invariance clusters of $\mathbb{I}$. Then there must be some set of values $\mathbf{c}_1, \mathbf{c}_2 \in \mathcal{D}_{\mathbf{C}}$ for some intervariable cluster $\mathbf{C} \in \mathbb{C}$ such that $\mathbf{c}_a$ and $\mathbf{c}_b$ are in the same cluster in one of $\mathbb{D}$ and $\mathbb{D}'$ but not in the other. Assume without loss of generality that they are in the same cluster in $\mathbb{D}$. The presence of the two values in the same cluster would imply (by Def. 8) that there exists a sequence $\mathbf{c}_1 = \mathbf{c}_a, \mathbf{c}_2, \mathbf{c}_3, \ldots, \mathbf{c}_N = \mathbf{c}_b$ such that for each $\ell \in \{1, \ldots, N-1\}$, there exists $g_{\mathbf{C}_i}^k$ and some $\phi_k \in \mathcal{D}_{\phi_k}$ such that either $g_{\mathbf{C}_i}^k(\mathbf{c}_\ell, \phi_k) = \mathbf{c}_{\ell+1}$ or $g_{\mathbf{C}_i}^k(\mathbf{c}_{\ell+1}, \phi_k) = \mathbf{c}_\ell$. If this is true, then $\mathbb{D}'$ is not a maximal invariance cluster of $\mathbb{I}$. Otherwise, $\mathbb{D}$ is not a maximal invariance cluster of $\mathbb{I}$. $\square$

One interesting consequence of this result is that there is a unique set of intravariable clusters that is most optimal given an SCM specification.

**Proposition 3** (Most Optimal Clusters). *Given an SCM $\mathcal{M}$, there is a unique set of intravariable clusters $\mathbb{D}^*$ such that the AIC is not violated, and all coarser clusters do violate the AIC. Moreover, these clusters are the maximal invariance clusters of structural functions $\mathbb{I}$, where $\mathbb{I} = \{g_{\mathbf{C}_i}^* : \mathbf{C}_i \in \mathbb{C}\}$ such that for any two $\mathbf{c}_1, \mathbf{c}_2 \in \mathbb{D}_{\mathbf{C}_i}$, $\mathbf{c}_1 = g_{\mathbf{C}_i}^*(\mathbf{c}_2, \phi_{\mathbf{C}_i})$ if and only if Eq. 1 holds.*

*Proof.* We first note that any set of clusters other than $\mathbb{D}^*$ that is not finer than $\mathbb{D}^*$ (including all coarser clusters) will violate the AIC. Consider an alternative cluster $\mathbb{D}'$, which must contain two values $\mathbf{c}_1, \mathbf{c}_2 \in \mathcal{D}_{\mathbf{C}}$ for some $\mathbf{C} \in \mathbb{C}$ that are in the same intravariable cluster in $\mathbb{D}'$ but not in $\mathbb{D}$.

However, this implies that two values for which Eq. 1 do not hold are clustered together, which violates the AIC by definition.

The uniqueness of $\mathbb{D}^*$ is guaranteed by Prop. 2, completing the proof. $\qquad\square$

This result is interesting as it implies, in a sense, a lower bound on the size of possible clusters that do not violate the AIC. Even with the most precise set of structural invariances, there are limitations based on the complexity of the SCM functions.

### D.3  OTHER RELATED WORKS

This paper makes a contribution in the direction of leveraging state-of-the-art representation learning techniques in causal contexts, using the theory of causal abstraction inference. This is not to be confused with *disentangled causal representation learning* (von Kügelgen et al., 2021; Shen et al., 2022; Brehmer et al., 2022; Varici et al., 2023; Ahuja et al., 2023; Squires et al., 2023; Wendong et al., 2023; Wang and Jordan, 2024; Zhang et al., 2024; Li et al., 2024), which is a well-studied subtopic of causal representation learning (Schölkopf* et al., 2021). The goal of such works is to discover high-level causal variables from available data where the mapping between data and variables are not immediately clear due to entanglement. Given the underspecification of such a challenging task, such works often require assumptions to avoid identifiability issues, including assuming availability of high-level variable labels, working in parametric spaces, or having the ability to perform interventions. In contrast, this paper works in the setting where the high-level variables are understood to be a constructive abstraction of the low-level variables in the data, and transformations are based on invariance information. There is no disentanglement of causal variables required.

We note that the concept of inter- and intravariable clusters is not to be confused with inter- and intraclass scatter (Vasilescu, 2024). While inter- and intravariable clusters describe the relationships between multiple variables and their values, inter- and intraclass scatter describes the variance comparing features with a label. Nonetheless, maximizing the ratio of interclass scatter to intraclass scatter may be helpful for learning invariant representations in cases where the causal structure involves several features pointing to a single important label.

While this work considers the most fundamental form of the AIC in Def. 5, there may be relaxed definitions that are easier to achieve and verifiable from data. An example is Chalupka et al. (2015), which shows that, in a confounded image recognition setting, one can achieve a set of clusters that almost always satisfies an interventional version of the AIC provided that it satisfies the observational version. This is described in more detail in Xia and Bareinboim (2024, App. D.2).

