# OpenReview forum: "Learning Invariances for Causal Abstraction Inference"
_ICLR.cc/2026/Conference — Submitted to ICLR 2026_

### Official Review · Reviewer_E4LW · 2025-10-30

**Soundness:** 2
**Presentation:** 2
**Contribution:** 2
**Rating:** 4
**Confidence:** 3

**Summary:**

The paper connects invariance learning to causal abstraction inference. It argues that invariances can help learn low-dimensional representations that satisfy the abstract invariance condition (AIC). The authors propose using contrastive learning to achieve this and test it on two small examples — a voting toy setup and a semi-synthetic pneumonia dataset.

**Strengths:**

- The paper provides a detailed background on causal abstraction and invariance concepts.
- There is an effort to make the connection between invariance and causal abstraction
- Experiments are easy to follow

**Weaknesses:**

- The method is basically a standard contrastive learning setup applied to causal abstraction — not much novelty in the learning approach.
- Experiments are very limited and “toyish,” offering little real empirical insight.
- The paper spends a lot on preliminaries but does not deliver a strong new idea or technical depth.

To conclude, I don t think that the takeaway using Contastive Learning to learn invariance is novel.

**Questions:**

Please address the points i have raised as weaknesses.

---

> ### Author Response · Authors · 2025-11-18
> **Rebuttal to Reviewer E4LW**
>
> We thank the reviewer for the comments and feedback. Please find our response below.
>
> > The method is basically a standard contrastive learning setup applied to causal abstraction — not much novelty in the learning approach.
>
> > The paper spends a lot on preliminaries but does not deliver a strong new idea or technical depth.
>
> > To conclude, I don t think that the takeaway using Contastive Learning to learn invariance is novel.
>
> **Response**: Thank you for the thoughtful feedback. We believe the concerns stem from a misunderstanding of the primary contribution of the paper and appreciate the opportunity to clarify. Our goal is not to introduce a new contrastive learning method or to claim novelty in how invariances are learned. Rather, the central contribution is the first general framework for performing causal abstraction inference in high-dimensional settings, overcoming the obstruction posed by the AIC.
>
> Section 2 provides the theoretical foundation that identifies when and how invariances can be leveraged to guarantee the AIC, enabling causal inference across abstractions even under arbitrary graphs, unobserved confounders, and general interventional or counterfactual queries. Prior approaches to causal abstraction inference could not scale beyond hand-crafted or low-dimensional settings. The novelty, therefore, lies in the causal abstraction inference framework, not in contrastive learning itself. Section 3 presents one illustrative instantiation using contrastive learning, but our theory applies to any representation-learning approach capable of incorporating the relevant invariances.
>
> > Experiments are very limited and “toyish,” offering little real empirical insight.
>
> **Response**: Regarding the experiments: their purpose is to validate the theoretical predictions from Sec. 2, rather than to position our method within broad empirical benchmarks. Causal abstraction inference currently lacks established high-dimensional benchmarks, precisely because existing methods do not scale to these regimes. Section 4.1 (the voting example) illustrates exactly how violating the AIC leads to incorrect causal inferences and how our framework resolves the issue. Section 4.2 (the pneumonia setting) shows that invariance-based representations substantially improve causal inference accuracy in high-dimensional image data, scenarios where previous causal abstraction methods cannot operate. Developing standardized benchmark suites is an important future direction, but it is a later stage of work that builds on the conceptual foundations established here.

---

> > ### Comment · Area_Chair_CepN · 2025-11-28
> >
> > Dear Reviewer,
> >
> > Please make sure you read the authors' response and engage with them in the discussion before the end of the discussion period on **Dec 03 '25 09:00 PM UTC**. This is a hard deadline.
> >
> > Thank you for supporting quality peer review at ICLR.
> >
> > AC

---

### Official Review · Reviewer_39mc · 2025-10-31

**Soundness:** 2
**Presentation:** 2
**Contribution:** 2
**Rating:** 2
**Confidence:** 3

**Summary:**

The paper focuses on learning causal abstractions, which are compressed representations of a causal system that preserve the ability to answer causal queries (interventions, counterfactuals) correctly. The authors propose to learn invariances that implicitly satisfy the Abstract Invariance Condition (AIC), which is a well-known requirement that ensures that high-level interventions correspond to consistent low-level interventions. Experiments are conducted on a synthetic voting dataset and real-world X-ray images, demonstrating the improvement of the proposed contrastive approach.

**Strengths:**

1. **Novel Theoretical Integration.** The paper provides a rigorous link between structural invariances and the Abstract Invariance Condition (AIC) in causal abstraction inference. It establishes that invariance functions can be leveraged to construct lower-dimensional representations that still satisfy the strict AIC, while providing an interpretable objective with sound theoretical guarantees.

2. **Practical Empirical Validation.** The improvements are consistent across embedding sizes and baselines.

**Weaknesses:**

1. **Strong Assumptions** The approach critically assumes known invariance information (Def. 7). As acknowledged in the discussion, this assumption is rarely realistic outside synthetic or well-controlled domains. Without it, the method cannot guarantee AIC satisfaction, limiting real-world applicability.

2. **Limited Baseline Evaluations**. The experiments focus on small-scale synthetic setups and a single real-world dataset (low-resolution chest X-rays), comparing the constructive and original versions of RNCM. There is no comparison with other existing causal representation learning approaches. Notably, "Self-Supervised Learning with Data Augmentations Provably Isolates Content from Style", which also extracts invariant content using contrastive learning.

3. **Contrastive Learning Practicality.** The contrastive training setup (Eq. 3) assumes sufficient batch diversity and correct pairing of equivalent samples. In practice, as noted, violations of these conditions could severely hinder convergence or lead to degenerate clusters. The work does not quantify the sensitivity of the results to these violations.

4. **Simplified Graphical Assumptions.** The approach relies on a predefined C-DAG, which can further restrict its real-world usability.

**Questions:**

1. **Scalability.** How does the proposed method scale to high-resolution or high-dimensional data?

2. **Invariances.**  Can the invariance discovery be automated rather than assumed? What happens if the assumed or discovered invariance set is partially wrong or incomplete?

---

> ### Author Response · Authors · 2025-11-18
> **Rebuttal to Reviewer 39mc (1/2)**
>
> We thank the reviewer for the comments and feedback. Please find our response below.
>
> > **Strong Assumptions** The approach critically assumes known invariance information (Def. 7). As acknowledged in the discussion, this assumption is rarely realistic outside synthetic or well-controlled domains. Without it, the method cannot guarantee AIC satisfaction, limiting real-world applicability.
>
> **Response**: We respectfully disagree with the reviewer that the assumption of invariant information is rarely realistic outside synthetic or well-controlled domains. The precise reason we chose this assumption is that it is prevalent across a large majority of applications in computer vision and other applied machine learning fields. The contrastive learning setup we use to demonstrate the approach's real-world applicability is based on SimCLR by Chen et al. (2020), which has over 20k citations. We use the same invariances leveraged by that paper, such as rotation, crop, etc., for images, and it is intuitive to see why such invariances hold for such datasets (e.g., a picture of a dog is still a dog even if rotated). The impossibility of guaranteeing AIC satisfaction without additional assumptions implies that some assumption must be made to make progress, and we chose to assume invariant information due to its wide real-world applications.
>
> >**Limited Baseline Evaluations.** The experiments focus on small-scale synthetic setups and a single real-world dataset (low-resolution chest X-rays), comparing the constructive and original versions of RNCM. There is no comparison with other existing causal representation learning approaches. Notably, "Self-Supervised Learning with Data Augmentations Provably Isolates Content from Style", which also extracts invariant content using contrastive learning.
>
> **Response**: Our work is the first to leverage invariances for causal abstraction inference, and we compare against appropriate baselines that solve the causal abstraction inference task without learning invariances. As we discuss in Appendix D.3, our work is not to be confused with works that claim to solve the “causal representation learning” task (including the cited paper mentioned by the reviewer, which is already discussed in D.3), as these works are solving a different task to disentangle causal variables from messy data (requiring a different set of assumptions). Our setting focuses on learning constructive abstractions of low-level variables, with no disentanglement required.
>
> > **Contrastive Learning Practicality.** The contrastive training setup (Eq. 3) assumes sufficient batch diversity and correct pairing of equivalent samples. In practice, as noted, violations of these conditions could severely hinder convergence or lead to degenerate clusters. The work does not quantify the sensitivity of the results to these violations.
>
> **Response**: The proof of Thm. 2 does require such assumptions, but these are justified before its proof in Appendix A.4. We note, in particular, that the sufficient batch diversity assumption is almost guaranteed to hold in high-dimensional settings, and for low-dimensional cases, we specifically provide Corol. 4, which quantifies the sensitivity of the results to violations of this assumption.
>
> > **Simplified Graphical Assumptions.** The approach relies on a predefined C-DAG, which can further restrict its real-world usability.
>
> **Response**: Graphical assumptions, such as the availability of the C-DAG is standard in causal inference tasks due to necessity. Without any causal assumptions, the Causal Hierarchy Theorem states that one can almost never make statements about the interventional or counterfactual levels using only observational data. Causal inference tasks are therefore studied under various sets of assumptions, and graphical assumptions are a flexible method of encoding causal constraints in general settings. The C-DAG can be thought of as a graphical abstraction of the causal diagram, and the availability of the C-DAG is, in fact, an even weaker assumption than in most causal inference tasks, where the entire causal diagram is assumed to be available.
>
> > **Scalability.** How does the proposed method scale to high-resolution or high-dimensional data?
>
> **Response**: As we demonstrate in the Pneumonia experiment in Sec. 4.2, our approach achieves much stronger performance in an image setting than previous methods. Practical challenges may arise in settings with much higher resolution, but these are engineering challenges related to the design of the NCM or the efficacy of the invariance learning approach. In general, we expect that improvements in these areas should also allow the methods presented by this paper to scale to higher-dimensional settings.

---

> ### Author Response · Authors · 2025-11-18
> **Rebuttal to Reviewer 39mc (2/2)**
>
> > **Invariances.** Can the invariance discovery be automated rather than assumed? What happens if the assumed or discovered invariance set is partially wrong or incomplete?
>
> **Response**: Yes, but this would require an additional assumption that the invariance discovery procedure is sound (i.e., the learned invariances satisfy Def. 7). In general, if the invariance set is smaller than expected, Corols. 2 and 3 guarantee that using it should not violate the AIC. However, if it is larger than expected (e.g., due to incorrect invariances), AIC violations may occur, potentially leading to arbitrarily bad causal inferences. It is therefore preferable to err on the side of caution and only add invariances that are almost sure to hold.
>
> We notice that the main weaknesses the reviewer highlighted concern the assumptions underlying our work. We believe that such criticisms are unwarranted, since theoretical progress is impossible in settings with no assumptions. So many research directions aim to understand the extent to which different assumptions provide further progress. Our assumptions are stated very transparently, and we justify each one by showing that it is very likely to hold across a variety of contexts. In light of this, we feel that the score provided is a bit harsh, and we hope that we have provided sufficient clarification in the rebuttal for the reviewer to raise their score. We are happy to add further clarifications if needed.

---

> > ### Comment · Area_Chair_CepN · 2025-11-28
> >
> > Dear Reviewer,
> >
> > Please make sure you read the authors' response and engage with them in the discussion before the end of the discussion period on **Dec 03 '25 09:00 PM UTC**. This is a hard deadline.
> >
> > Thank you for supporting quality peer review at ICLR.
> >
> > AC

---

### Official Review · Reviewer_tKCS · 2025-11-03

**Soundness:** 3
**Presentation:** 4
**Contribution:** 2
**Rating:** 6
**Confidence:** 3

**Summary:**

This paper proposes theorems and practical approaches to solve the causal abstraction inference problem, in which the abstract invariance condition (AIC) is a major restriction to be tackled. The theorem can be validated from toy datasets and chest-X ray medical image datasets.

**Strengths:**

+ The research problem and main focus are well introduced with substantial knowledgeable context such as the Figure 1 that clearly shows the problem setting of causal abstraction inference.

+ The proposed theorem-1 seems to be important, as it points out a new direction to enable SCM to satisfy AIC from the perspectives of the intervariable clusters and max invariance clusters.

+ A practical use case and empirical approach are provided to demonstrate the advantages of the proposed theorem. Specifically, a contrastive learning approach is found useful to learn invariances when training the existing RNCM model.

**Weaknesses:**

- The Introduction part does not present an overview of the proposed method, as well as how the method could achieve the stated merits as listed in the last paragraph on page-2.

- In the pneumonia experiment, although the chest-X ray images are from real-world applications, the invariances of translation, zooming, cropping, etc., are still synthetic or unrealistic in practice. It could be better to find more practice invariances.

- Technically speaking, the proposed contrastive learning method is nothing new, though this paper presents its advantages of learning the invariances.

**Questions:**

No additional questions.

---

> ### Author Response · Authors · 2025-11-18
> **Rebuttal to Reviewer tKCS**
>
> We thank the reviewer for the review and encouraging comments. Please find our responses below addressing the weaknesses.
>
> >The Introduction part does not present an overview of the proposed method, as well as how the method could achieve the stated merits as listed in the last paragraph on page-2.
>
> **Response**: Thank you for the suggestion. We will be happy to use the extra page to include a new figure cohesively summarizing the contributions of the work and the application pipeline.
>
> >In the pneumonia experiment, although the chest-X ray images are from real-world applications, the invariances of translation, zooming, cropping, etc., are still synthetic or unrealistic in practice. It could be better to find more practice invariances.
>
> **Response**: We chose to use the same invariances as in the SimCLR paper by Chen et al. (2020), which serves as the basis for our contrastive learning implementation. These invariances are widely accepted in the computer vision community as practical across many tasks and datasets, and it is intuitive to see why they may hold. Indeed, we could consider other invariances that are more complicated or perhaps even learn the invariances under some additional assumptions, but this is riskier, and incorrect assumptions could lead to AIC violations. The goal of this paper is to rely on the wisdom of existing research and provide an avenue for applying tools that are very effective at learning invariances in non-causal settings to causal settings.
>
> > Technically speaking, the proposed contrastive learning method is nothing new, though this paper presents its advantages of learning the invariances.
>
> **Response**: We do not change the contrastive learning technique itself; one of the novelties of our work is showing how it can be adapted to causal settings, with theoretical guarantees and a practical implementation via the RNCM. We note that while Sec. 3 focuses on the applications of contrastive learning specifically, the theory in Sec. 2 is general and can apply to any invariance learning method.

---

> > ### Comment · Area_Chair_CepN · 2025-11-28
> >
> > Dear Reviewer,
> >
> > Please make sure you read the authors' response and engage with them in the discussion before the end of the discussion period on **Dec 03 '25 09:00 PM UTC**. This is a hard deadline.
> >
> > Thank you for supporting quality peer review at ICLR.
> >
> > AC

---

### Official Review · Reviewer_CKjE · 2025-11-03

**Soundness:** 3
**Presentation:** 3
**Contribution:** 2
**Rating:** 4
**Confidence:** 3

**Summary:**

The paper presents a theoretical and algorithmic framework connecting invariance learning with causal abstraction inference. The approach presented aim at learning representations that preserve the Abstract Invariance Condition (AIC), which ensures that high-level abstractions preserve causal effects from their low-level mechanisms, and proves that structural invariances (e.g., rotations or permutations) define maximal invariance clusters that satisfy AIC while permitting dimensionality reduction (Theorem 1).
Building on this, the authors implement a contrastive learning approach within a Graph-Constrained Representational Neural Causal Model (GC-RNCM). By treating invariant transformations as positive pairs, they show that the resulting representations recover AIC-compliant clusters (Theorem 2).
Empirical validation is performed on synthetic (“Voting dataset”) and semi-synthetic (“Pneumonia dataset”) datasets show that these invariant embeddings improve causal-query estimation.

**Strengths:**

* **Good theoretical contribution.**
  The connection between structural invariances and causal abstractions is novel and rigorously established by Theorems 1–2 and Corollaries 1–3. These results are relevant to ground causal abstractions theory to practical contrastive learning algorithms. Moreover  the notion of maximal invariance clusters offers an intuitive, interpretable link between dimensionality reduction and causal faithfulness.


* **Presentation clarity.**
  The theory is presented in a clear and intuitive way: Examples (voting, rotated images) and diagrams (Figs. 3–6) build intuition and help the reader go through the theoretical statements. The definitions and notation are precise and consistent with existing causal-model semantics.

**Weaknesses:**

* **Gaps between theory and practice**: one of the main weaknesses of the paper are the gaps between the assumptions in the theoretical setting and real settings. While it is not expected to fully bridge these gaps in one paper, the risk is that the method developed only can work in very unrealistic and ideal settings. Below I detail these gaps:
   -    **Reliance on assumed invariances.** The framework presumes that a correct set of structural invariances is known a priori. The authors acknowledge this in the limitations section in the Appendix, but they do not analyze how approximation or misspecification of $\mathcal{I}$ affects AIC satisfaction or downstream causal inference. Since the end goal motivation is to apply the method in realistic domains, the lack of robustness or sensitivity analysis leaves this assumption hard to be verified in practice.
  -   **Unverifiable AIC in practice.**
  The appendix correctly notes that AIC cannot be empirically verified because the true structural mechanisms $f_V$ are unobserved. However, the paper does not explore measures or proxies (e.g., cross-intervention consistency) that could indicate when learned embeddings fail to preserve causal distinctions. For example, two low-level samples with distinct causal effects but similar invariant transformations, such as rotated X-ray images of different pathologies, could be mapped to the same embedding, yielding good predictive performance yet violating AIC. The issue is acknowledged but not addressed in depth.




* **Limited experimental scope** Experiments focus on two small-scale datasets (“Voting” and “Pneumonia”) and lack ablations over key parameters such as temperature, or invariance strength. The authors mention data limitations in Appendix B, but a broader empirical validation, especially on larger or multimodal settings, would be needed to demonstrate generality of method even when the assumptions hold just approximately.
    -    **Scalability and robustness not evaluated**.  The paper provides no discussion or empirical evidence on computational scaling with continuous invariance groups or higher-dimensional data. This limits again the applicability of the method.


* **Missing Related Work**
Several important directions are missing from the related work discussion:
     - **(1) Invariances and Identifiability in Causal Representation Learning.** The paper would benefit from connecting its theoretical contribution to recent works that explicitly link invariance principles with identifiability. For example:  [a] establishes that invariance constraints are both necessary and sufficient for identifying causal factors.  [b]: shows that grouping observed variables under structural constraints enables identifiability, conceptually akin to discovering invariant clusters. Despite not developing theory directly for causal abstractions, these works emphasize that structural or statistical invariances can serve as sufficient conditions for causal identifiability and need to be discussed.


     - **(2) Contrastive Learning for Disentanglement** While the paper leverages contrastive learning to enforce AIC-compliant abstractions, it omits prior works that exploit contrastive or hierarchical objectives to achieve disentanglement and invariance. For instance: [c]  proposes a contrastive manifold-projection loss that isolates independent latent factors, conceptually related to invariant causal clusters. [d] demonstrates that contrastive learning can recover latent factors consistent with the underlying generative process.[e] provides theoretical guarantees that contrastive learning with data augmentations separates invariant and variant components.

Beyond disentangled representations, hierarchical or multilevel contrastive frameworks offer direct analogies to multi-level causal abstraction and could serve as meaningful empirical baselines: [f] introduces a hierarchical contrastive objective across label levels, mirroring multi-tier causal abstraction. [g] employs hierarchical prototypes as higher-level anchors for contrastive learning, analogous to abstract variables summarizing lower-level clusters.  [h] learns hierarchical representations in hyperbolic space, providing an example of compositional abstraction akin to causal hierarchies.
Including or contrasting against such methods would help clarify how the proposed GC-RNCM framework differs from existing hierarchical or disentanglement-based contrastive approaches.


_[a] Yao et al. (2024). *Unifying Causal Representation Learning with the Invariance Principle.* ICLR 2025._

_[b] Morioka & Hyvärinen (2023). *Causal Representation Learning Made Identifiable by Grouping of Observational Variables.*_

_[c] Fumero et al. (2021). *Learning Disentangled Representations via Product Manifold Projection.* ICML 2021._

_[d] Zimmermann et al. (2021). *Contrastive Learning Inverts the Data-Generating Process.* NeurIPS 2021._

_[e] von Kügelgen et al. (2021). *Self-Supervised Learning with Data Augmentations Provably Isolates Content from Style.* NeurIPS 2021._

_[f] Zhang, Shu, et al. (2022). *Use All the Labels: A Hierarchical Multi-Label Contrastive Learning Framework.* CVPR 2022._

_[g] Li et al. (2021). *Prototypical Contrastive Learning of Unsupervised Representations.* ICLR 2021._

_[h] Pal, Avik, et al. (2025). *Compositional Entailment Learning for Hyperbolic Vision-Language Models.* ICLR 2025._



While I believe the theoretical contribution is of value, I also think that the issues above should be addressed or alternatively the scope of the paper should change accordingly and contributions on experiments and adaptability to practical settings should be tuned down.

**Questions:**

* Could the invariances be learned adaptively, e.g., through group-equivariant or augmentation-discovery architectures?
* How robust is performance to partially valid or noisy invariances?
* Can Theorem 2’s assumptions (“sufficient batch diversity”) be quantified empirically, e.g., through contrastive mutual-information bounds?

---

> ### Author Response · Authors · 2025-11-18
> **Rebuttal to Reviewer CKjE (1/2)**
>
> We thank the reviewer for the detailed review and constructive feedback. Please find our responses below.
>
> > **Reliance on assumed invariances.** The framework presumes that a correct set of structural invariances is known a priori. The authors acknowledge this in the limitations section in the Appendix, but they do not analyze how approximation or misspecification of $\mathcal{I}$ affects AIC satisfaction or downstream causal inference. Since the end goal motivation is to apply the method in realistic domains, the lack of robustness or sensitivity analysis leaves this assumption hard to be verified in practice.
>
> **Response**: We appreciate the concern. It may help to emphasize that this paper represents the first step beyond the impossibility barrier of Prop. 1. Prior to our work, no tractable set of assumptions was known under which the AIC could be satisfied at all. As a result, deeper robustness or sensitivity analysis is necessarily downstream: such analysis presupposes a baseline family of assumptions to vary or relax, and this paper is the first to identify such a family.
>
> We chose invariances because they are among the weakest and most commonly used structural assumptions in machine learning, and they are often far more testable in practice than full causal models. They therefore provide a natural starting point for applications where these properties are already well justified.
>
> Regarding robustness, our framework already provides practical flexibility: Corollaries 2 and 3 show that one can always adopt weaker invariances (i.e., finer partitions) without risking AIC violations. Practitioners who are uncertain about specific invariances can therefore use more conservative versions while retaining the guarantees. A more detailed sensitivity analysis is a natural direction for future work, but it logically follows from the foundational step established here.
>
> > **Unverifiable AIC in practice.** The appendix correctly notes that AIC cannot be empirically verified because the true structural mechanisms $f_V$ are unobserved. However, the paper does not explore measures or proxies (e.g., cross-intervention consistency) that could indicate when learned embeddings fail to preserve causal distinctions. For example, two low-level samples with distinct causal effects but similar invariant transformations, such as rotated X-ray images of different pathologies, could be mapped to the same embedding, yielding good predictive performance yet violating AIC. The issue is acknowledged but not addressed in depth.
>
> **Response**: Indeed, we do not provide methods to verify if the AIC holds, and we discuss related work in the paper that resolves this issue by studying relaxed variants of the AIC that are testable. In this work, we do not need to verify the AIC due to the invariance assumption discussed earlier. We note that the issue and example raised by the reviewer is an exact case that is covered by the invariance assumption. Specifically, if a rotation follows Def. 7, then Thm. 1 guarantees that it will not violate the AIC, and therefore, we can trust that despite mapping the two X-ray images to the same embedding, the downstream causal inferences will not be affected.
>
> > **Limited experimental scope** Experiments focus on two small-scale datasets (“Voting” and “Pneumonia”) and lack ablations over key parameters such as temperature, or invariance strength. The authors mention data limitations in Appendix B, but a broader empirical validation, especially on larger or multimodal settings, would be needed to demonstrate generality of method even when the assumptions hold just approximately.
>
> **Response**: We thank the reviewer for the suggestions. We will add ablation tests relating to certain parameters, such as temperature. Regarding invariance strength and approximate assumptions, we note that this is already shown in our results through variations in embedding sizes, as embeddings that are too small will force violations of the AIC, and this worsens as they become even smaller. Multimodal causal inference is interesting but poorly understood, so scaling to such settings is out of the scope of our paper.
>
> > **Scalability and robustness not evaluated.** The paper provides no discussion or empirical evidence on computational scaling with continuous invariance groups or higher-dimensional data. This limits again the applicability of the method.
>
> **Response**: We are not sure if we have understood the reviewer correctly in this statement. We believe the X-ray image dataset is an example of a higher-dimensional setting, and the invariances we use are largely the same as those in the SimCLR paper, which are continuous. We respectfully ask if the reviewer could clarify the issue further.

---

> ### Author Response · Authors · 2025-11-18
> **Rebuttal to Reviewer CKjE (2/2)**
>
> > **(1) Invariances and Identifiability in Causal Representation Learning.** The paper would benefit from connecting its theoretical contribution to recent works that explicitly link invariance principles with identifiability. For example: [a] establishes that invariance constraints are both necessary and sufficient for identifying causal factors. [b]: shows that grouping observed variables under structural constraints enables identifiability, conceptually akin to discovering invariant clusters. Despite not developing theory directly for causal abstractions, these works emphasize that structural or statistical invariances can serve as sufficient conditions for causal identifiability and need to be discussed.
>
> **Response**: Thank you for the suggestions, we will include citations for these works. We note in Appendix D.3 that the major difference between our work and works in the “causal representation learning” umbrella is that in our work, the high-level variables are understood to be a constructive abstraction of the data. In contrast, in other works, the mapping between data and high-level variables is less clear and often entangled, and therefore, they require many other types of assumptions to make sound inferences. Our work does not require disentanglement.
>
> > **(2) Contrastive Learning for Disentanglement** While the paper leverages contrastive learning to enforce AIC-compliant abstractions, it omits prior works that exploit contrastive or hierarchical objectives to achieve disentanglement and invariance. For instance: [c] proposes a contrastive manifold-projection loss that isolates independent latent factors, conceptually related to invariant causal clusters. [d] demonstrates that contrastive learning can recover latent factors consistent with the underlying generative process.[e] provides theoretical guarantees that contrastive learning with data augmentations separates invariant and variant components.
>
> **Response**: Thank you for the suggestions, we will include citations for these works. While these works provide valuable insight into the usage of contrastive learning, we note that the lack of causal context leaves their application to our work unclear. Nonetheless, we believe that the framework presented in our work provides a clear avenue for future research on how to incorporate these powerful insights into causal inference.
>
> > Could the invariances be learned adaptively, e.g., through group-equivariant or augmentation-discovery architectures?
>
> **Response**: Yes, it is possible, but there would need to be an underlying assumption that such invariances are aligned with the generative process of the data (i.e., following Def. 7). Otherwise, it is not guaranteed that invariances learned from observational data will mean anything on the causal level (similar to issues resulting from the Causal Hierarchy Theorem).
>
> > How robust is performance to partially valid or noisy invariances?
>
> **Response**: In theory, having weaker invariances is not problematic due to Corols. 2 and 3, but having stronger (false) invariances can result in arbitrarily bad performance if the AIC is violated. In practice, the severity of the performance drop can depend on the degree of AIC violation, as we demonstrate in the scaling embedding sizes.
>
> > Can Theorem 2’s assumptions (“sufficient batch diversity”) be quantified empirically, e.g., through contrastive mutual-information bounds?
>
> **Response**: This is an interesting idea, thank you. As it stands, the “sufficient batch diversity” assumption is only problematic in low-dimensional settings, as we discuss in Corol. 4 in Appendix A.4. In such cases, the batch diversity assumption can be directly quantified numerically and does not require empirical evaluation. In high-dimensional settings that are more commonly found in practice, the assumption is almost guaranteed to hold, but such bounds could be interesting to study nonetheless.

---

> > ### Comment · Area_Chair_CepN · 2025-11-28
> >
> > Dear Reviewer,
> >
> > Please make sure you read the authors' response and engage with them in the discussion before the end of the discussion period on **Dec 03 '25 09:00 PM UTC**. This is a hard deadline.
> >
> > Thank you for supporting quality peer review at ICLR.
> >
> > AC

---

### Meta-Review · Area_Chair_gsVJ · 2026-01-02

**Summary:**

The reviewers’ main concerns focus on the gap between the paper’s strong theoretical contribution and its practical and empirical support. In particular, several reviewers question the realism and robustness of the assumptions.

Reviewers also point out that the learning component relies on contrastive learning techniques, and that the novelty lies primarily in the theoretical integration with causal abstraction rather than in the algorithmic method itself, with limited comparison.

The experimental evaluation is considered too limited in scope to substantiate claims about generality, scalability, and practical utility, as it is restricted to small synthetic examples and a single semi-synthetic image dataset.

**Reviewer Concerns:**

Assumptions and Scope of Applicability

Several reviewers raise concerns about the strength and realism of the assumptions underlying the proposed framework, e.g., the approach critically relies on the availability of correct structural invariances. In particular, reviewers question their realism and verifiability in practical applications, especially in complex, real-world domains where invariances may be only approximate, partially valid, or unknown.

In the rebuttal, the authors argue that some form of assumption is unavoidable due to impossibility results. While this justification is conceptually sound, it does not fully address the reviewers’ concerns regarding robustness to misspecified or incomplete invariances, nor does it clarify how failures of these assumptions would affect AIC satisfaction or downstream causal inferences. As a result, the gap between the conditional theoretical guarantees and their reliability in realistic settings remains an important open issue.

Use of Contrastive Learning

Reviewers also question the role and novelty of contrastive learning in the proposed framework. While the paper uses contrastive objectives to recover invariance-based clusters, several reviewers note that contrastive learning for enforcing invariances or disentangling latent factors is already well studied. From this perspective, the learning component itself is seen as technically standard, and its added value for causal abstraction inference is not fully distinguished from prior work on invariant or disentangled representations.

In the rebuttal, the authors clarify that their main contribution is not a new contrastive method but rather the theoretical integration of invariances with causal abstraction via the AIC, and that contrastive learning is only one illustrative instantiation of this idea. This clarification helps position the contribution more precisely. However, the lack of empirical comparison to related contrastive or disentanglement-based approaches leaves some uncertainty about how much the proposed framework differs in practice from existing methods that also leverage invariances.

Experimental Evaluation and Practical Utility

A central concern across multiple reviews is the limited scope of the experimental evaluation. The experiments do not explore more complex, heterogeneous, or genuinely multimodal settings. Reviewers therefore question whether the empirical results are sufficient to support the generality and practical relevance of the framework.

The rebuttal clarifies that the experiments are primarily intended to validate the theoretical claims rather than to provide broad empirical benchmarks, and notes data and benchmark limitations in this area. While this explanation is reasonable, it does not fully address the concern that the current empirical evidence may be narrow to substantiate claims about scalability, robustness, or applicability beyond carefully controlled scenarios.

Overall Assessment

Overall, while the rebuttal clarifies the conceptual motivations and theoretical positioning of the work, it does not fully resolve the reviewers’ concerns regarding the strength and realism of the assumptions, the practical distinctiveness of the contrastive learning component, and the sufficiency of the experimental validation. Additional analysis of robustness to assumption violations, clearer differentiation from existing invariant or disentanglement-based approaches, and broader empirical evaluation, especially in more complex or heterogeneous settings, would be necessary to more convincingly support the claimed generality and practical utility of the framework.

**Reviewer Scores:**

Reviewer CKjE: The reviewer raised substantial concerns about the strength and realism of the assumptions, the unverifiability of the AIC in practice, and the limited experimental scope. While the rebuttal clarified the theoretical positioning of the work, it did not fully resolve these concerns. I therefore do not expect this reviewer’s score to increase and anticipate it would remain largely unchanged.

Reviewer 39mc: This reviewer appreciated the theoretical integration but was concerned about strong assumptions, limited baselines, robustness, and scalability. The rebuttal did not substantially mitigate the concerns about applicability and experimental coverage. I do not expect a positive change in this reviewer’s assessment.

Reviewer E4LW: This reviewer was primarily concerned about novelty and the limited empirical evaluation, but was otherwise not strongly negative and explicitly indicated openness to acceptance. The rebuttal clarified the contribution and helped address misunderstandings about the intended scope. I therefore expect this reviewer might slightly increase the score, but not dramatically change their overall assessment.

---

### Decision · Program_Chairs · 2026-01-26

Reject